

# Frequent Ultrafine Particle Formation and Growth in the Canadian Arctic Marine Environment

Douglas B. Collins[1], Julia Burkart[1], Rachel Y.-W. Chang[2], Martine Lizotte[3], Aude Boivin-Rioux[4], Marjolaine Blais[4], Emma L. Mungall[1], Matthew Boyer[2], Victoria E. Irish[5], Guillaume Massé[3], Daniel Kunkel[6], Jean-Éric Tremblay[3], Tim Papakyriakou[7], Allan K. Bertram[5], Heiko Bozem[6], Michel Gosselin[4], Maurice Levasseur[3], Jonathan P.D. Abbatt[1]

[1]Department of Chemistry, University of Toronto, Toronto, ON, M5S 3H6, Canada
[2]Department of Physics and Atmospheric Science, Dalhousie University, Halifax, NS, B3H 4R2, Canada
[3]Québec-Océan, Département de Biologie, Université Laval, Québec, QC, G1V 0A6, Canada
[4]Institut des sciences de la mer Rimouski, Université du Québec à Rimouski, Rimouski, QC, G5L 3A1, Canada
[5]Department of Chemistry, University of British Columbia, Vancouver, BC, V6T 1Z1, Canada
[6]Johannes Gutenberg University of Mainz, Institute of Atmospheric Physics, 55099 Mainz, Germany
[7]Center for Earth Observation Science, University of Manitoba, Winnipeg, MB, R3T 2N2, Canada

*Correspondence to*: Douglas B. Collins (douglas.collins@utoronto.ca)

**Abstract.** The source strength and capability of aerosol particles in the Arctic to act as cloud condensation nuclei have important implications for understanding the indirect aerosol-cloud effect within the polar climate system. It has been shown in several Arctic regions that ultrafine particle (UFP) formation and growth is a key contributor to aerosol number concentrations during the summer. This study uses aerosol number size distribution measurements from ship-board measurement expeditions aboard the research icebreaker CCGS *Amundsen* in the summers of 2014 and 2016 throughout the Canadian Arctic to gain a deeper understanding of the drivers of UFP formation and growth within this marine boundary layer. UFP number concentrations (diameter > 4 nm) in the range of $10^1 \sim 10^4$ cm$^{-3}$ were observed across the two seasons, with concentrations greater than $10^3$ cm$^{-3}$ occurring more frequently in 2016. Higher concentrations in 2016 were associated with UFP formation and growth, with events occurring on 41% of days, while events were only observed on 6% of days in 2014. Assessment of relevant parameters for aerosol nucleation showed that the median condensation sink in this region was approximately 1.2 h$^{-1}$ in 2016 and 2.2 h$^{-1}$ in 2014, which lie at the lower end of ranges observed at even the most remote stations reported in the literature. Apparent growth rates of all observed events in both expeditions averaged $4.3 \pm 4.1$ nm h$^{-1}$, in general agreement with other recent studies at similar latitudes. Higher solar radiation, lower cloud fractions, and lower sea ice concentrations combined with differences in the developmental stage and activity of marine microbial communities within the Canadian Arctic were documented and help explain differences between the aerosol measurements made during the 2014 and 2016 expeditions. These findings help to motivate further studies of biosphere-atmosphere interactions within the Arctic marine environment to explain the production of UFP and their growth to sizes relevant for cloud droplet activation.



# 1    Introduction

Polar regions have been experiencing more rapid climate changes than the mid-latitudes (AMAP, 2012; Vaughan et al., 2013), prompting enhanced research activities in both the Arctic and Antarctic. Arctic sea ice extent, for instance, has been decreasing throughout the modern period of satellite measurements (Simmonds, 2015; Stroeve et al., 2012). The expansion of open water

in the Arctic could lead to an increase in anthropogenic pollution sources due to increased access to shipping routes (Law and Stohl, 2007) along with possible enhancements in natural ocean-atmosphere exchange processes (Browse et al., 2014; Levasseur, 2013), some of which could influence concentrations of precursors for aerosol formation (Becagli et al., 2016; Rempillo et al., 2011; Sharma et al., 2012; Shaw et al., 2010). In general, changes in natural aerosol particle production and subsequent atmospheric processes contribute importantly to uncertainty in aerosol-cloud-climate interactions (Carslaw et al.,

2013; Tsigaridis et al., 2013).

With important effects on cloud formation through their role as cloud condensation nuclei (CCN), understanding the sources, sinks, and physicochemical properties of atmospheric aerosol particles is important for predicting cloud microphysics and aerosol indirect effects on climate (Carslaw et al., 2013; Ramanathan et al., 2001). Low level clouds in the Arctic have a

substantial impact on both the shortwave and longwave radiation balance depending on the season, with cooling effects on climate dominating in the summer (Intrieri et al., 2002b; Lubin and Vogelmann, 2010; Walsh and Chapman, 1998). In general, the extent of Arctic low level liquid clouds reaches a maximum in the warm season, especially over the oceans (Cesana et al., 2012; Intrieri et al., 2002a) where the sensitivity of radiative forcing due to the aerosol indirect effect is strongest owing to large differences in surface and cloud albedo.

During Arctic summer, the relatively high cloud fraction and greater degree of in-cloud and below-cloud aerosol scavenging play an important role in reducing ambient aerosol concentrations compared with winter and spring (Browse et al., 2012; Croft et al., 2016b), in addition to evidence for weaker long-range transport of pollutants from mid-latitudes in summer (Stohl, 2006). Typically, CCN-active aerosol particles with critical supersaturations relevant to marine stratiform liquid cloud formation are

thought to have diameters ($d_\mathrm{p}$) similar to or greater than 100 nm (depending on composition), corresponding roughly with so-called 'accumulation mode' aerosol (Hegg et al., 2012). In environments with low CCN concentrations, the sensitivity of cloud droplet number concentrations to changes in CCN are typically stronger than cases with larger CCN concentrations (Ramanathan et al., 2001). Seasonal trends in aerosol size distribution measurements have consistently shown that sub-100 nm particles are more abundant than accumulation mode particles throughout the Arctic during summer in the boundary layer

(e.g., Asmi et al., 2016; Croft et al., 2016b; Heintzenberg et al., 2015; Nguyen et al., 2016; Tunved et al., 2013); such small particles would require greater water vapour supersaturations to nucleate cloud droplets than accumulation mode particles. The seasonal shift in the size distribution toward smaller particles in summer is therefore consistent with the suggestion that





cloud microphysical changes can be somewhat insensitive to ground-level aerosol concentrations in the Arctic summer (Garrett et al., 2004).

New results challenge that view. In particular, a recent chemical transport model study, validated by aircraft measurements in the Canadian Arctic, suggests a relatively strong sensitivity between cloud droplet number concentration and aerosol concentration in the summertime Arctic (Moore et al., 2013). In-situ measurements of liquid clouds in the Canadian Arctic performed in summer 2014, coincident with some of the data presented within the present study, indicated that accumulation mode aerosol particles were often acting as the principal cloud droplet nuclei in lower level clouds (cloud base < 200 m). However, in higher altitude clouds (cloud base > 200 m), the minimum diameter particle that was CCN-active was substantially less than 100 nm, suggesting that water vapour supersaturations in clouds with higher base heights were greater than expected (Leaitch et al., 2016); Arctic clouds could respond to changes in sub-100 nm aerosol concentrations as long as the concentrations of particles larger than 100 nm were sufficiently low. Since the concentration of sub-100 nm aerosol particles in summer is typically greater than the accumulation mode number concentration (e.g., Asmi et al., 2016; Heintzenberg et al., 2015; Leaitch et al., 2013; Nguyen et al., 2016; Tunved et al., 2013), their production is of great importance to cloud properties in Arctic summer provided that water vapour supersaturations can exceed critical supersaturations for such small particles, as shown by Leaitch et al. (2016). The nature of the sub-100 nm aerosol source, its strength or production flux, and the physicochemical properties of the particles in question are all key parameters to constrain.

The relatively high concentrations of sub-100 nm particles in the Arctic during summer have been mostly associated with aerosol nucleation (Asmi et al., 2016; Chang et al., 2011; Giamarelou et al., 2016; Heintzenberg et al., 2015; Karl et al., 2012; Kolesar et al., 2017; Leaitch et al., 2013; Shaw, 1989; Ström et al., 2009; Tunved et al., 2013; Wiedensohler et al., 1996), although contributions from primary marine aerosol (Leck and Bigg, 2005) and the evaporation of fogs has been suggested as well (Leck and Bigg, 2010). Ultrafine particles (UFP), defined in this study as aerosol particles with $d_p$ = 4-20 nm, have a vertical profile maximum in the Arctic boundary layer both in the Canadian Archipelago and in the vicinity of Svalbard (Burkart et al., 2016; Engvall et al., 2008b) suggesting a source of UFP at or near the Earth's surface. While the source regions and precursor components are still a topic of active research, studies have shown that ammonia ($NH_3$) and dimethyl sulphide (DMS) are associated with the formation and growth of UFP in the Arctic (Chang et al., 2011; Croft et al., 2016a; Ferek et al., 1995; Ghahremaninezhad et al., 2016; Giamarelou et al., 2016; Heintzenberg and Leck, 1994; Leaitch et al., 2013; Leck and Persson, 1996), with likely contributions from organic material during particle growth (Willis et al., 2016). Biogenic iodine compounds have also been implicated in UFP formation in temperate coastal zones (Mäkelä et al., 2002; O'Dowd et al., 2002) and in the Arctic near eastern Greenland (Allan et al., 2015). Overall, most studies have suggested marine or coastal sources for nucleation and growth precursors in the Arctic (Burkart et al., 2016; Chang et al., 2011; Croft et al., 2016a; Heintzenberg et al., 2015; Sharma et al., 2012; Willis et al., 2016), although a recent study in Alaska has shown that UFP formation and


growth can also be strongly influenced by anthropogenic emissions associated with oil production activities (Kolesar et al., 2017), at least for regions proximal to such activities.

This study uses aerosol measurements from two Arctic summertime campaigns aboard the icebreaker CCGS *Amundsen* in
2014 and 2016 to investigate the frequency of, and factors contributing to, UFP formation and particle growth toward CCN-relevant sizes within the marine boundary layer. Detailed size distribution analyses of UFP formation and growth events in Arctic locations have only recently been published (Asmi et al., 2016; Kolesar et al., 2017; Nguyen et al., 2016), and the present study is the first to present such an analysis strictly within the Arctic marine environment. The study is unique in its wide and consistent spatial coverage across two expeditions, the similar seasonal timing of the expeditions, and the large
number of co-sampled atmospheric and oceanic parameters, permitting a wide range of environmental conditions to be considered. Specifically, these data are distinct from the more numerous Arctic UFP data sets that have been gathered at long-term monitoring stations located on land.

The general characteristics of the atmospheric aerosol measured during each campaign will be described in detail and
comparisons between expeditions will be made in light of their environmental similarities and differences. Further, in order to constrain the range of conditions that may limit UFP formation and growth in the Arctic marine environment, meteorological and oceanic conditions were investigated throughout each of the expeditions. The goal of this study is to define the frequency and characteristics of UFP formation and growth within the remote Canadian Arctic, characterize the environmental factors that are associated with UFP formation in the Arctic marine boundary layer, and to provide broad motivation to more
comprehensively understand secondary aerosol precursors in the marine/coastal environment.

## 2    Measurement Methods

### 2.1    Atmospheric Aerosol Measurements

Measurements of ambient atmospheric aerosol were conducted between 15 July – 12 August 2014 and 20 July – 23 August 2016 aboard the research icebreaker CCGS *Amundsen*, operating within the Canadian Arctic as part of a multi-year research
project, NETCARE (Network on Climate and Aerosols: Addressing Key Uncertainties in Remote Canadian Environments). The cruise track for each of the two field campaigns is provided in Figure 1.

Ambient concentrations of aerosol with $d_p > 4$ nm were measured using an ultrafine condensation particle counter (UCPC; TSI, Inc. Model 3776), operating with an inlet flow rate of 1.5 L min$^{-1}$. While the nominal lower size limit for detectable
particles for this instrument was specified by the manufacturer at 2.5 nm, diffusional losses of particles in the tubing (stainless steel, 4.57 mm inner diameter) increased the practical lower size limit to approximately 4 nm. Concentrations were sampled at 1 Hz, and were subsequently averaged to time bins of coarser resolution for calculating various size-resolved aerosol metrics



in conjunction with other data products. Number size distributions of particles between 10 – 430 nm were measured using a scanning mobility particle sizer (SMPS; TSI, Inc. Model 3080/3787) operating with a sample flow rate of 0.6 L min$^{-1}$ and a sheath air flow rate of 6.0 L min$^{-1}$. SMPS and UCPC sampled from the starboard side of the ship's foredeck, approximately 5 meters aft of the bow and approximately 7 meters above the sea surface. Number size distributions of particles with diameter

between 0.54 – 20 μm were measured with an aerodynamic particle sizer (APS; TSI, Inc. Model 3021) from atop the ship's bridge using a louvered inlet designed for total suspended particle transmission and a straight vertical stainless steel tube (16.56 mm inner diameter) coupled directly to the inlet of the APS (total flow rate 5 L min$^{-1}$).

The influence of ship exhaust was excluded from the 2016 data by applying a wind direction filter to the data in post-processing

with an acceptance angle of 60° to port and 90° to starboard of the ship's heading. Extension of the acceptance angle to 90° on the starboard side is related to the position of the sampling inlet near the starboard edge of the ship on the foredeck; winds arriving at the sampling inlet from the starboard side, directly perpendicular to the ship's heading were free from contamination. Within the 2014 dataset, high frequency fluctuations in aerosol concentration, caused by intermittently sampling ship exhaust, remained in the data record after filtering with the aforementioned wind direction criteria. The influence

of ship exhaust was excluded from the 2014 data by first discarding all data points with SMPS concentrations > 5000 cm$^{-3}$ and UCPC concentrations > 10,000 cm$^{-3}$. Further, the SMPS data were filtered for time periods where particle concentrations increased by more than 200% of the median of the preceding 10 consecutive valid measurements, given that the elevated concentration only persisted for less than 3 data points. Periods in which SMPS total concentrations varied continuously by more than 200% were also excluded, as they indicated sustained sampling of transient ship exhaust plumes. The difference in

data filtering method between the two years arises from different relative wind direction frequency distributions, as well as the rate at which the wind direction changed relative to the ship's compass heading. Using only the wind direction filter for the 2014 data would have preserved its main features, but also would have included some undesirable ship exhaust signal.

Total aerosol concentration and number size distribution data were used to calculate various metrics to describe the aerosol

characteristics succinctly. The number concentration of particles with $d_p$ = 4-20 nm (N$_{4-20}$) has been used previously as a metric for UFP (e.g., Burkart et al., 2016; Leaitch et al., 2013), and was obtained using Eq. 1:

$$N_{4-20} = N_{UCPC} - \int_{20}^{430} N(d_p)\mathrm{d}d_p \tag{1}$$

where $N_{UCPC}$ is the total concentration of particles measured by the UCPC, $N(d_p)$ is the number size distribution obtained from the SMPS, and the limits of the integral are in units of nanometres. In addition to N$_{4-20}$, the quantity N$_{4-10}$ was calculated and

represents the difference between the total concentration measured by the UCPC (lower limit $d_p$ = 4 nm) and the SMPS (lower limit $d_p$ = 10 nm). The aerosol condensation sink (CS) was calculated using both SMPS and APS data according to Eq. 2 and 3 (Dal Maso et al., 2002):

$$CS = 2\pi D \sum_i \beta_i d_{pi} N_i \tag{2}$$





where $i$ represents a size bin of particles having diameter $d_{pi}$ and number concentration $N_i$. D is the diffusion coefficient, given a value of $7 \times 10^{-6}$ m$^2$ s$^{-1}$ corresponding with an oxidized organic molecule (Tang et al., 2015). The Fuchs-Sutugin transition regime correction factor $\beta_i$ was computed in each size bin using Eq. 3:

$$\beta_i = \frac{1+Kn}{1+\left(\frac{4}{3\alpha}+0.337\right)Kn+\frac{4}{3\alpha}Kn^2} \tag{3}$$

where $\alpha$ is the dimensionless sticking coefficient ($\alpha = 1$), Kn is the (dimensionless) Knudsen number, $2\lambda/d_p$, and $\lambda$ is the mean free path of vapour molecules ($\lambda = 65$ nm).

## 2.2   Atmospheric State Measurements and Air Mass History Modeling

Meteorological state variables (e.g., wind direction, wind speed, temperature, and relative humidity) were measured from a tower on the ship's foredeck. Relative humidity (RH) and air temperature were measured using a shielded probe (Vaisala™
HMP45C212 in 2014 and HMP155A in 2016) at approximately 14.5 m and 16.3 m above sea level during respectively the 2014 and 2016 cruises. Wind direction and speed were measured using a wind monitor (RM Young™ 05103-10), positioned at 16.2 m and 17.6 m above sea level in 2014 and 2016. Sensors were scanned every 2 s and saved as 2 min averages to a micrologger (Campbell Scientific™, model CR3000). Platform relative wind was post-processed to true wind following Smith et al. (1999). Navigation data (ship position, speed over ground, course over ground and heading) necessary for the conversion
were available from the ship's position and orientation system (Applanix POS MV™ V4). Periods when the tower sensors were serviced, or when the platform relative wind were beyond ± 90° from the ship's bow were screened from the data set. Downwelling shortwave solar radiation was measured using a pyranometer (Eppley model PSP) secured to a purpose built platform atop the ship's bridge. The sensor was scanned every second and stored as 2 min averages on a micrologger (Campbell Scientific™ model CR1000).

Air mass histories were computed using the Flexible Particle Dispersion (FLEXPART) model driven by meteorological analysis data from the European Centre for Medium-Range Weather Forecast (ECMWF). The analysis data is used with a horizontal grid spacing of 0.25 degrees in longitude and latitude with 137 hybrid-pressure levels in the vertical. Tracers are inert, non-interacting particles that are released from the ship's position every six hours. Tracers were continuously released
for one hour between 10-20 m above sea level. FLEXPART was run in backward mode and model output is given as the spatially-resolved potential emission sensitivity (PES) (or residence time) of the tracer particles over a particular location, available every three hours. PES represents the amount of time that an air mass may be influenced by emissions from a given location in space and time. For analysis in this study, PES has been vertically-integrated from the surface up to 10 km above mean sea level and time-integrated for 5 days (120 hours) leading up to the release time.



## 2.3    Satellite Observations

Satellite retrievals were used to estimate cloud cover and sea ice concentration in the study region. Cloud cover was estimated using the monthly Level 3 MODIS-Aqua mean liquid cloud fraction product (http://modis-atmos.gsfc.nasa.gov/MOD08_M3/). Data were obtained for all latitudes from 30-90 °N at 1° spatial resolution for the months August 2014 and August 2016

through the United States National Aeronautics and Space Administration (NASA) Goddard Space Flight Center (GSFC) Giovanni interface (https://giovanni.sci.gsfc.nasa.gov/giovanni/). Average values for cloud fraction were calculated within a box bounded by the limits 68-82 °N and 55-110 °W, inclusive. Sea ice was assessed from the National Snow and Ice Data Center (NSIDC) Defense Meteorological Satellite Program Special Sensor Microwave Imager Sounder (SSMIS) Daily Polar Gridded Sea Ice Concentrations (http://nsidc.org/data/nsidc-0081) (Maslanik and Stroeve, 1999). Data were retrieved for 15

July – 31 August in both 2014 and 2016. Both cloud fraction and sea ice concentration data and were visualized using the NASA Panoply Data Viewer (https://www.giss.nasa.gov/tools/panoply/).

## 2.4    Oceanic Measurements

Sea surface temperature was measured through the ship's Inboard Shiptrack Water System using a Seabird/Seapoint measurement system. Samples for the concentration of DMS in seawater ($DMS_{sw}$) were collected from the sea surface via

Niskin-type bottles (OceanTest Equipment) in 2014 (depth = 0-0.5 m) and using the underway seawater sampler aboard the CCGS *Amundsen* in 2016 (depth = 4 m), allowing for greater spatial resolution. In 2014, discrete samples of $DMS_{sw}$ were quantified aboard the ship within 2 h of collection by gas chromatography (GC; Varian 3800) following purging with helium (flow rate of 50 mL min$^{-1}$) and cryotrapping in liquid nitrogen as described by Lizotte et al. (2012). In 2016, $DMS_{sw}$ was quantified by gas chromatography/mass spectrometry (GC/MS; Agilent 7890A/5975C) coupled to a permeable gas membrane

module (PermSelect®) with 7500 cm$^2$ of exchange surface and a multimodal cryogenic trap inlet. Automated measurements of $DMS_{SW}$ were taken and logged every 10 min and linked with the ship's global positioning system through custom-designed LabView software (StudioBods, Inc.). A constant flow (0.2 mL min$^{-1}$) of a standard solution of $d_3$-DMS and $d_6$-DMS in the seawater line upstream (constant flow of 100 mL min$^{-1}$) of the permeable membrane allowed for continuous dual internal isotopic calibration.

Samples for dissolved organic carbon (DOC), chlorophyll *a* (chl *a*), and primary production (PP) were collected via Niskin-type bottle or a bucket at the sea surface (depth < 4 m). DOC was determined using a high-temperature combustion Shimadzu TOC-V$_{CPN}$ Total Organic Carbon Analyzer, as described in detail by Mungall et al. (2017) Chl *a* concentrations, an index of phytoplankton biomass, was measured using a 10-AU Turner Designs fluorometer following the acidification method of

Parsons et al. (1984) PP was estimated using the $^{14}$C-assimilation method during 24 h simulated *in situ* deck incubations (Ardyna et al., 2011; Knap et al., 1996). Chl *a* and PP were both measured on particles retained on Whatman GF/F filters (nominal pore size of 0.7 um) and 5 um Nuclepore polycarbonate membrane filters.





Vertically-resolved seawater samples for nitrate ($NO_3^-$) determinations were collected with Niskin bottles attached to a CTD-Rosette water sampler. The concentrations of $NO_2^-$ (nitrite) and $NO_3^-+NO_2^-$ were measured on fresh samples with a *Bran+Luebbe Auto-Analyzer 3* using adaptations of the colorimetric methods of Grasshoff et al.(2009), with an analytical

detection limit of 0.03 uM. Nitrate was obtained by difference.

Expedition-wide analysis of environmental parameters measured aboard CCGS *Amundsen* was performed by calculating the number of observations that fell within prescribed bins of each quantity (e.g., $DMS_{sw}$, solar radiation, RH). Each frequency distribution was then normalized such that the sum of the frequencies across all bins was unity, similar to a probability

distribution. For satellite products, averages and average differences in sea ice and cloud fraction were obtained by vectorising the spatially-resolved matrix of data before computing the mean. Grid points with missing data were disregarded from statistical analysis. This data analysis approach precluded spatial patterns from influencing statistics, so that all points within the study region were treated with equal weight. All calculations were performed using standard MATLAB® functionality (Version R2016b; The MathWorks, Inc.).

**3    Results and Discussion**

Measurements of aerosol total number concentration and size-resolved number concentration from 32 days in 2014 and 34 days in 2016 were analysed to investigate factors contributing to formation and growth of UFP. In the following sections, general observations of ultrafine particles in the Canadian Arctic are reported and discussed in the context of surrounding environmental conditions.

**3.1    Ultrafine Particle Concentrations**

Summertime UFP concentrations ($N_{4-20}$) in the Canadian Arctic marine environment had a range of about 3 orders of magnitude (approx. $10^1$-$10^4$ cm$^{-3}$) during 2014 and 2016 observation periods. Histograms of $N_{4-20}$ measured aboard the CCGS *Amundsen* are given in Figure 2. Most commonly, $N_{4-20}$ was between 50-100 cm$^{-3}$ for both 2014 and 2016, but differences are evident when examining the frequency at which higher concentrations were observed. Summer 2016 exhibited a greater frequency of

observations with $N_{4-20}$ > 1000 cm$^{-3}$, with 5.1% of the 5-minute average measurements of $N_{4-20}$ in excess of 2000 cm$^{-3}$ (<1% of observations exceeded 2000 cm$^{-3}$ during the 2014 cruise). The absolute maximum 5-minute average $N_{4-20}$ concentration observed in 2016 was 9350 cm$^{-3}$, whereas the maximum 5-minute average concentration in 2014 was 5521 cm$^{-3}$. It should be noted that maximum 1 second concentrations observed by the UCPC in 2016 were in excess of $10^4$ cm$^{-3}$ and were observed during UFP formation events, but were short lived in the time series (smooth features lasting 5-15 min) due to various factors

including ship movement.





Elevated $N_{4-20}$ concentrations were observed in similar locations during 2014 and 2016. In 2014, $N_{4-20}$ was in excess of 1000 $cm^{-3}$ on fewer isolated occasions than in 2016 (Figures 2 and 3). Instances of elevated $N_{4-20}$ in 2014 were located along the northeast coast of Baffin Island, near the eastern extent of Lancaster Sound (near Pond Inlet and Bylot Island), in Upper Baffin Bay near the coast of Greenland, and within Nares Strait (between Ellesmere Island and Greenland). On two of these occasions, size distribution measurements indicate that UFP formation and/or growth was occurring, as the temporal profile of the size distributions resembled that of typical aerosol nucleation events observed in more heavily studied continental regions (Kulmala et al., 2014). Occasions in which $N_{4-20} > 1000$ $cm^{-3}$ in 2014 were localized to a few regions: the northeast coast of Baffin Island, around Bylot Island (near Pond Inlet), in Upper Baffin Bay near the coast of Greenland, within the Nares Strait, and in the Franklin Strait/Queen Maude Gulf region within the Northwest Passage (south of Resolute Bay). With perhaps an exception of the Franklin Strait/Queen Maude Gulf region, for each of the locations in which elevated $N_{4-20}$ was observed in 2014, generally similar trends in concentration were observed in 2016. In most cases, high $N_{4-20}$ concentrations were associated with aerosol nucleation and/or particle growth, as discussed in detail below. The significant degree of spatial consistency in $N_{4-20}$ between 2014 and 2016 supports the notion that the production process for UFP may be consistent and/or geographically localized within the Canadian Arctic.

## 3.2 Ultrafine Particle Formation Events

### 3.2.1 Temporal Characteristics and General Description

Associated with the higher frequency of elevated $N_{4-20}$ concentrations in 2016 compared with 2014, the number of instances in which UFP were observed to form and grow in diameter was also greater in 2016. A UFP formation event is defined here as a temporally contiguous period (scale of hours) in which $N_{4-20}$ was elevated, SMPS measurements indicate that particles were observed in the smallest size bins, and the slope in modal size of particles over time is positive. A particle growth event is defined as a temporally contiguous period (scale of hours-days) in which SMPS measurements indicate that particles were steadily increasing in diameter (but not necessarily starting from the smallest measurable sizes in the SMPS); growth events may include one dominant population (mode) of particles, or multiple distinct modes growing simultaneously. Regions in which UFP formation and/or growth events were observed are indicated by black boxes in Figure 3. While there were many more instances of UFP formation/growth events in 2016, the two locations in which UFP formation was observed in 2014 were roughly co-located with events observed in 2016.

Two example time periods from the 2016 campaign are shown in Figure 4. It should be noted that in both example cases, the ship was moving ahead at approximately 6 m $s^{-1}$, which could affect apparent particle growth rates and/or the appearance/disappearance of events. Both events show noticeable increases in $N_{4-10}$ in the beginning with its fractional contribution to $N_{total}$ decreasing with time. In addition, substantial concentrations of particles in the smallest bins of the SMPS



were observed at the beginning of the event, providing strong evidence that the events observed can be characterized as aerosol nucleation (Kulmala et al., 2014; Zhang et al., 2012).

In Figure 4a, a 'gap' in the event where particle concentrations drop abruptly to less than 1000 cm$^{-3}$, was observed during
which the ship moved through a bank of low fog. Particle concentrations likely were reduced due to fog scavenging processes (i.e., nucleation scavenging, impaction scavenging) and/or local depletion of aerosol growth precursor gases by uptake to fog droplets (Pruppacher and Klett, 2010). After the ship travelled through the fog (approx. 11 km), characteristics of the event resumed in a similarly abrupt way, indicating that the fog patch was contained within the broader spatial extent of the nucleation/growth event. Also, the peak concentration of particles is notable, as total particle concentrations in excess of 12,000
cm$^{-3}$ (1-minute average; $d_p > 4$ nm) were observed during this event.  This strong nucleation burst (along with other nucleation/growth events immediately following the event, shown in Figure 6) was located in the Nares Strait, where an event was also observed in 2014, suggesting the possibility of a substantial, consistent source of precursors in the region. Ship-board trace gas measurements in the same region of interest during the 2014 CCGS *Amundsen* expedition have shown elevated values of NH$_{3(g)}$ (Wentworth et al., 2016), gas-phase DMS (Mungall et al., 2016), and evidence of an ocean source for other volatile
organic compounds (Mungall et al., 2017) that may act as precursors for growth.

Previous detailed studies in the Arctic have shown that the occurrence of UFP formation and growth events was associated with biogenic sulphur compounds like DMS, which has a substantial oceanic source, and its lower vapour pressure oxidation products, methanesulphonic acid (MSA) and sulphuric acid (H$_2$SO$_4$) (Chang et al., 2011; Ghahremaninezhad et al., 2016;
Leaitch et al., 2013; Quinn et al., 2002; Rempillo et al., 2011). Other studies have shown that the duration of contact that the air mass had with open water along its backward trajectory was positively correlated with the occurrence of UFP formation (Asmi et al., 2016; Heintzenberg et al., 2015) and aerosol biogenic sulphur concentrations (Sharma et al., 2012). Recently, Kolesar et al. (2017) segregated air mass backward trajectories to show that the frequency of nucleation events observed at a station near the Beaufort Sea was greater for coastal influenced air masses than those that were associated mainly with open
water. Coastal areas can be associated with elevated emissions of various trace gases, including iodine species (Mäkelä et al., 2002), NH$_3$ (Croft et al., 2016a; Wentworth et al., 2016), and/or reduced sulphur compounds (Bates and Cline, 1985; Leck and Rodhe, 1991; Turner et al., 1988). Coastal zones have also been shown to influence aerosol nucleation in locations outside the Arctic (O'Dowd et al., 2002; Weber et al., 1998). Air mass histories shown in Figure 5 for four example events from 2016 indicate that within 5 days prior to each event, air was contained within the Arctic environment and largely over the ocean or
sea ice, although coastal influence was notable as well, especially for the 31 July 2016 case (impacted by coastlines within Baffin Bay; Figure 5a) and to some extent for the 20 August 2016 case (coastal influence along northern Canada and Alaska; Figure 5d). Despite N$_{4-20}$ > 1000 cm$^{-3}$ near Pond Inlet in 2016 (Figure 3), neither UFP formation nor particle growth was observed in this region; the 5-day air mass history indicates that air sampled during this period was impacted strongly by Northern Canada and Alaska (Figure S1). Overall, when considering locations within or near the Canadian Archipelago, the



air masses that have been 'marine influenced' will likely also be impacted by coastal zones. The largely marine and coastal influence for air masses with notable UFP formation and growth observed aboard CCGS *Amundsen* in 2014 and 2016 highlight the importance of oceanic sources for secondary aerosol precursors in the Arctic.

Another example period is shown in Figure 4b. The beginning of this time period (until approximately 20 Aug 16 15:00 UTC) was characterized by average 18.2 m s$^{-1}$ apparent wind and an elevated sea state, leading to substantial local sea spray production; UFP formation was observed despite these conditions, as the condensation sink was approx. $4\times10^{-4}$ s$^{-1}$ (1.4 h$^{-1}$), which is on the low end of ranges that were observed in the boreal forest, a North Atlantic coastal site, and on the north coast of Alaska where UFP formation and growth was also observed (Dal Maso et al., 2002; Kolesar et al., 2017; Westervelt et al.,

2013) (see also Section 3.2.4). The following day, a subsequent new particle formation event was observed, causing the existing populations of particles to grow in parallel; the two modes of particles did not appear to be combined as a result of growth in the smaller mode. This type of successive particle growth was observed on multiple occasions in 2016 and may be an important mechanism for the growth of newly-formed particles to sizes sufficient for CCN activity. In both cases, growth proceeded during daylight hours and ceased in periods of darkness (solar radiation < 50 W m$^{-2}$ from 02:00 UTC to the end of Figure 4a

and from 01:30 UTC 20 Aug – 12:30 UTC 21 Aug in Figure 4b); particle size distributions were temporally consistent through the night on each of the example occasions.

### 3.2.2    Frequency of Events

The overall temporal profiles of SMPS size distributions (Figure 6) show substantial differences between the overall concentrations, frequency of high particle concentration events, and the frequency with which particle growth was observed

between the NETCARE 2014 and 2016 campaigns. In 2016, temporally contiguous periods can be seen in which particle concentrations were high (saturating the scale in Figure 6) and tend toward growth of modal diameters over time. Growth from newly formed particles up to diameters approaching 100 nm can be seen occurring over the scale of days. It is important to note that the ship was moving during the vast majority of the campaign so the appearance and disappearance of events may be due to ship movement or changes in meteorological conditions. Overall, though, it is clear that conditions must have been

more favourable for ultrafine particle formation and growth in 2016 than in 2014. Beyond the visible differences in size distributions shown in Figure 6, UFP formation and growth was documented on 14 occasions in 2016 (41% of observation days) and on 2 occasions in 2014 (6% of observation days). Events tended to be observed on consecutive days, with defined gaps between groups of events; since the ship was moving most of the time, temporal grouping translated to regional grouping of nucleation and/or growth events.



### 3.2.3 Particle Growth Rates

Particle growth is an important factor driving the concentration of CCN in the Arctic summer (Willis et al., 2016). Growth of particles observed in this study was quantified by fitting the temporal trend in modal diameter to a line, where the slope is equal to the growth rate (Dal Maso et al., 2005). Since the measurements reported here were made on a moving measurement platform, growth rates should be seen as 'apparent' growth rates ($GR_{app}$), since ship movement and air mass advection could have influenced the magnitude of the rate observed (Kivekäs et al., 2016). $GR_{app}$ in the Canadian Arctic varied widely, from 0.2-15.3 nm h$^{-1}$ ($\bar{x}$ = 4.3 nm h$^{-1}$, σ = 4.1 nm h$^{-1}$; Figure 7). Still, the range of $GR_{app}$ reported here agrees generally with those observed during July and August at land-based coastal Arctic sites. Utqiaġvik, Alaska averaged growth rates of 3.6 and 5.0 nm h$^{-1}$ for July and August (2008 – 2015), respectively (Kolesar et al., 2017), and measurements in Tiksi, Russia indicated a range of monthly-averaged rates of 2.6-4.8 nm h$^{-1}$ for the July/August time period (2010-2014), depending on how events were classified and segregated (Asmi et al., 2016). A higher altitude study at Summit, Greenland (3200 m above sea level) during summer 2007 showed four growth events with average GR between 0.09-0.3 nm h$^{-1}$ (Ziemba et al., 2010), possibly a result of the distance from sources of condensable vapour. As pointed out in the reports of both the Utqiaġvik and Tiksi studies, growth rates observed in the Arctic were similar to those measured in lower latitude environments (Asmi et al., 2016; Kolesar et al., 2017).

Within specific events summarized in Figure 7 (starting date/time coded in event identifiers; format: '*GMMDD_hhmm*'), some evidence for decreasing growth rate with increasing diameter can be observed (e.g., event on 8/14/2016: G0814_1452, G0814_1532, G0814_2119), although modal size and growth rate were not well correlated overall within the 2016 data ($r^2$ = 0.02). While some authors have shown that linear trends in growth can be used to evaluate growth rates (Dal Maso et al., 2005; Kolesar et al., 2017), in some cases reported in the present study – particularly those producing large number concentrations – the growth rate slowed over time as particles grew larger. When this was clearly occurring, the event was split into sub-sections to best classify the apparent growth rates observed in this environment, leading to multiple events with different growth rates and size ranges in Figure 7. Event periods with growth rates higher than 3 nm h$^{-1}$ were often associated with particle diameters smaller than 50 nm, but a global relationship did not exist (Figure S2). The range of growth rates of the smallest particles measured in this study is in general agreement with observations at Tiksi, Russia, where some events showed slow growth rates (< 1 nm h$^{-1}$), while other events indicated faster growth and larger ultimate particle sizes (Asmi et al., 2016). Simultaneous growth of two or more modes was observed in a variety of cases (e.g., Figure 4b) and will be discussed in detail in a separate manuscript.

### 3.2.4 Condensation Sink

The condensation sink (CS) is represented by the first order rate constant for the removal of condensable vapour via condensation on existing aerosol particles and is a key parameter when assessing atmospheric aerosol nucleation (Dal Maso et




al., 2002; Kulmala et al., 2001; Pirjola et al., 1999). Aerosol particles will nucleate when CS is low, as the energy barrier to nucleation is higher than that for condensation on existing particle surfaces. CS measured during the 2014 and 2016 expeditions was within the range of 0.2-10 $h^{-1}$, with median values of 2.15 $h^{-1}$ in 2014 and 1.21 $h^{-1}$ in 2016. During all growth events catalogued in Figure 7, CS remained on the lower end of the range observed throughout the study, but still covered an order

of magnitude range of values (0.3-3.45 $h^{-1}$). CS was uncorrelated with $GR_{app}$ ($r^2 = 0.007$) and did not appear to have major influence on the occurrence of growth events in the Canadian Arctic during the study periods (Figure S3). With that noted, total CS was particularly low (<1 $h^{-1}$) leading up to and during the beginning of both substantial UFP formation events detailed in Figure 4.

In order to better understand the sizes of particles that accommodate condensable vapour during each expedition, CS was re-calculated using different groups of size bins in Equation 2, specifically, just the size bins that were measured by the SMPS ($CS_{SMPS}$; $d_p < 430$ nm) and just the SMPS size bins for $d_p < 100$ nm ($CS_{100}$). For 95% of the observations in 2016, greater than 65% of CS was derived from particles with $d_p < 430$ nm (Figure 8a). Coarse particle sources like sea spray, therefore, did not appear to have substantial influence on the degree to which conditions were conducive to nucleation in the summertime

Canadian Arctic. In fact, the UFP formation event shown in Figure 4b was initially observed simultaneously with primary marine aerosol production that brought the contribution of particles with $d_p > 500$ nm to CS up to ~50%, but the absolute value of CS was still at the low value of approximately 1.4 $h^{-1}$, as discussed previously. The $CS_{100}$, on the other hand, was often a substantial fraction of the total condensation sink in 2016 (Figure 8b), with $CS_{100}/CS_{TOT} > 0.5$ for ~25% of observations. This indicates that the nucleating and/or growing particles were an important contributor to total CS during that season, and may

account for the prevalence of UFP growth observed especially often in 2016, as the capacity for larger particles to accommodate condensable material was low and variable (Figure 8b). Particles with $d_p < 100$ nm contributed less to total CS in 2014, as UFP formation was much less frequent; only one main peak in Figure 8b can be observed.

Comparing the normalized cumulative distribution of total CS for the 2014 and 2016 study periods with published median

values from continental locations (Donahue et al., 2016; Westervelt et al., 2013), the Canadian Arctic marine environment exhibited a median CS that was a factor of 3-6 lower than that observed in Hyytiälä, Finland (median CS of 6.3 $h^{-1}$), a heavily studied remote boreal forest location where nucleation is commonly observed (Kulmala et al., 2014). The range in CS from the present study overlaps with the lower end of the ranges observed during the summer at Utqiaġvik, Alaska (Kolesar et al., 2017), Zeppelin station near Ny Ålesund, Svalbard (Giamarelou et al., 2016), and Mace Head, Ireland (Dal Maso et al., 2002)

(Figure 8c). Prior studies in the Arctic and elsewhere have noted that low CS was important for UFP formation to occur in the Arctic (Chang et al., 2011; Engvall et al., 2008a; Leaitch et al., 2013) although the importance of low CS was often contrasted with the highly polluted 'Arctic Haze' period in the spring season. The present study suggests that relatively low CS values in the summertime Canadian Arctic marine boundary layer are more common and widespread than other locales – particularly those at lower latitudes – and therefore may not be a factor that directly limits the formation of UFP in this region.





### 3.3 Environmental Conditions

In order to provide context to the observations from each of the two expeditions, a suite of atmospheric and oceanic conditions was compiled and is shown in Figures 9 – 11. In this study, only expedition-wide analysis of the data, rather than temporal correlations or finer scale analysis, have been presented in light of the complex relationship expected between basic
environmental parameters and the formation of UFP in the atmosphere

### 3.3.1 Atmospheric Conditions

Ambient air temperature and relative humidity frequency distributions illustrate basic similarities between the two expeditions, owing to the consistent time of year and geographical extent of the measurements, although some slight differences exist. The 2016 expedition experienced a wider range of ambient temperatures (Figure 9b) with the lowest temperatures (below 0 °C)
occurring at the northernmost extent of the cruise track (81° N latitude), and the warmest temperatures (10-15 °C) occurring mainly in the Queen Maude Gulf region (68° N latitude) at the southwest extent of the study area. Relative humidity was generally lower in 2016 (Figure 9d), suggesting that fogs and low clouds may have been less prevalent. This is further supported by the stronger peak in the frequency distribution of daily maximum solar radiation around 400 W m$^{-2}$ in 2016 (Figure 9a, dashed lines). Studies have suggested that removal of particles in the Arctic summer is strongly coupled to wet
scavenging (Browse et al., 2012; Croft et al., 2016b), so the suggestion of more common fogs and low clouds based on the relative humidity and solar radiation data would support the fact that aerosol concentrations were generally lower in 2014 with UFP formation and growth occurring less often.

Cloudiness in the Arctic for each of the two expeditions was more directly compared using monthly MODIS liquid cloud
fraction retrievals, using August data from both years. While average liquid cloud fraction in the study region between 2014 and 2016 (within the box in Figure 10) indicates the lack of an overall difference in cloudiness between 2014 and 2016, ($\bar{x}_{2014}$ = 0.28, $\bar{x}_{2016}$ = 0.28), the difference map of cloud fraction indicates that spatial differences existed, with a range in the per-grid square differences between 2016 and 2014 of -0.3 to 0.3 depending on location (Figure S3). The geographic distribution of cloud fraction and its difference between 2016 and 2014 within the study region differs to a degree that may help explain
differences observed in aerosol concentration and UFP formation/growth, with the largest negative values in Figure 10c occurring throughout Upper Baffin Bay, Nares Strait, and Queen Maude Gulf where UFP formation/growth was observed frequently in 2016.

Solar radiation is known to drive photochemical reactions in the gas phase which produce low vapour pressure compounds
that contribute to particle formation and growth (Kulmala et al., 2014; Zhang et al., 2012). Recently, solar radiation has also been suggested to influence the flux of volatile organic compounds to the gas phase in the marine environment through photochemical reactions at the air-sea interface (Brüggemann et al., 2017; Chiu et al., 2017). The average incoming shortwave




solar radiation measured during each growth event was always lower than the maximum daily solar radiation (Figure 7a), since UFP formation was commonly initiated in the afternoon. $GR_{app}$ showed moderate positive correlation with solar radiation ($r^2$ = 0.30), but only if the highest rates ($GR_{app} > 10$ nm h$^{-1}$) were excluded from consideration (Figure S3). Since the Arctic boundary layer in summer is typically stable (Tjernström et al., 2012), it is likely that high solar radiation enhanced the

concentrations of condensable materials in the gas phase, which were concentrated within the boundary layer. A recent report of vertical profiles of UFP in this region showed a maximum in the boundary layer (Burkart et al., 2016), suggesting a source that is associated with the Earth's surface.

In summary, cloud cover can be associated with both wet scavenging of aerosol along with simultaneous dimming of the

planetary boundary layer and surface. While wet scavenging is an important removal mechanism for aerosol in the Arctic summer (Browse et al., 2012; Croft et al., 2016b) and the resulting low CS is important for priming the environment for aerosol nucleation to occur (Dal Maso et al., 2002; Kreidenweis et al., 1991; Pirjola et al., 1999), low solar radiation at the surface resulting from cloud cover may also cause a reduction in the source of precursors for UFP formation and growth. Solar radiation measured aboard CCGS *Amundsen* in 2016 was greater than in 2014, and when combined with a generally low CS and typically

stable Arctic marine boundary layer, one finds that the conditions were overall more conducive to UFP formation in 2016 than in 2014. The source strengths of gas-phase precursors and reactive species, which are generally not well understood in the marine environment, are key remaining factors for explaining inter-annual differences and regional variability in UFP formation in the Arctic marine environment.

### 3.3.2 Oceanic Conditions

Sea ice concentrations in the Canadian Arctic were lower in 2016 than 2014, especially during August. Averaged over the study period, sea ice concentration was approximately 13% lower in 2016 throughout the Canadian Arctic as a whole relative to 2014, albeit with substantial spatial variability (Figure 11). The relative daily average differences in sea ice concentration between 2016 and 2014 (Figure S4) were between -6% and +3% in mid-July, with a steep shift to relative differences between -13% to -25% during August. Open water was exposed in 2016 throughout Lancaster Sound, Barrow Strait, Peel Sound, and

Queen Maude Gulf (locations west of ~90 °W longitude). Sea ice coverage was greater in 2016 along the coast of Baffin Island and within the Nares Strait. Ice along Baffin Island in 2016 was discontinuous first-year ice with many open leads and melt ponds mixed with transported icebergs. In Nares Strait, iceberg production by glaciers was a large contributor to sea ice, with open water between them. Despite increased sea ice in two locations where UFP formation/growth was observed, the presence of ice was discontinuous and marine sources of UFP precursors would not have been hindered to a major degree (Loose et al.,

2014). In addition to remotely sensed sea ice coverage, the difference in the amount of time that CCGS *Amundsen* spent in sea ice between the two expeditions can also be noted in the histograms of sea surface temperature (SST; Figure 9c), where a distinct local maximum in normalized frequency of measurements just below 0 °C is notable only in 2014.


It has long been thought that secondary aerosol formation and its influence on CCN concentrations in the marine environment is associated with biological activity in the ocean, involving the atmospheric chemistry of both organic and inorganic volatile precursors (e.g., Charlson et al., 1987; Facchini et al., 2008; Fu et al., 2013; O'Dowd et al., 2002; O'Dowd and de Leeuw, 2007), although some assessments have tempered expectations of climate feedback mechanisms to exist in tropical and

temperate zones (Heintzenberg et al., 2004; Pirjola et al., 2000; Quinn and Bates, 2011). The prevalence of strong associations between UFP and air mass exposure to open ocean in the Arctic, however, suggests that UFP formation and growth may be more strongly coupled to oceanic biological activity given more generally favourable atmospheric conditions in summer (Heintzenberg et al., 2015; Leaitch et al., 2013; Leck and Bigg, 2010; Rempillo et al., 2011; Sharma et al., 2012).

Frequency distributions of different metrics that have been associated with marine microbiological activity are shown in Figure 9e-h: $DMS_{sw}$, DOC, chl $a$, and PP. While each of these metrics has a different relationship to the broad notion of 'marine microbiological activity', taken together, they may provide valuable information on the general state of the marine biological system during each expedition. Three of the four aforementioned metrics exhibited broadly similar characteristics between the two cruises, with subtle, but discernible, differences. DOC concentrations were nearly identical between the two summers; it

is possible that DOC concentrations in this region are driven to a substantial degree by physical processes (e.g., ocean mixing) and/or inputs from terrigenous sources (Dittmar and Kattner, 2003; Hansell et al., 2009) rather than just marine biological processes. While $DMS_{sw}$ concentrations have a similar modal concentration in the frequency distribution, a larger fraction of the measurements showed concentrations greater than 1 nmol $L^{-1}$ in 2016 compared with 2014 (Figure 9e). In most locations, chl $a$, PP, and nitrate skewed lower in 2016 than 2014 (Figure 9g-i; see also Figure S6), suggesting that the phytoplankton

production season was more advanced and well into the post-bloom phase. This difference is consistent with the greater contribution of small phytoplankton to chl $a$ and may be related to the lower sea ice concentration in 2016. $DMS_{sw}$ production results from the enzymatic cleavage of the cellular osmolyte dimethylsulphoniopropionate (DMSP) often found in microalgae at different intracellular levels in relation to phytoplankton species composition (Keller et al., 1989) and oxidative stress (Stefels et al., 2007; Sunda et al., 2002). The breakdown process from DMSP to DMS is performed by certain phytoplankton

and is widespread among bacterioplankton (Stefels et al., 2007). The latter micro-organisms may play a greater role in processes such as DMS production when PP declines towards the later stages of a phytoplankton bloom (Azam et al., 1983). Changes in the chemical and physicochemical properties of marine aerosol have been associated with declining phytoplankton abundance; it is thought that dynamic ecosystem processes, including hydrolytic enzyme interactions with organic matter, are important to aerosol production and composition (Lee et al., 2015; O'Dowd et al., 2015; Wang et al., 2015). If such a

biogeochemical regime were to exist during the 2016 cruise, which could be explained by lower PP, nitrate, and chl $a$ with higher $DMS_{sw}$, it could also be associated with enhanced emissions of trace gases by microbial communities (Carpenter et al., 2012; Schulz and Dickschat, 2007; Shaw et al., 2010), potentially including those that could act as precursors to UFP formation and/or aerosol growth. While the full suite of gases emitted from biologically active oceans is not well understood (although a dependence on community composition has been shown (e.g., Colomb et al., 2008)), the general understanding that trace gas





production can be enhanced by certain biological interactions and productivity (e.g., Leck and Rodhe, 1991; Mäkelä et al., 2002; Shaw et al., 2010) points to an association with increased UFP formation and/or growth, as observed in summer 2016.

## 4    Conclusions

This study presents detailed number size distribution measurements of aerosol particles in the Canadian Arctic marine boundary layer during two different summer seasons, 2014 and 2016. This unique dataset highlights the wide spatial extent of UFP in the summertime Arctic, and a range in particle number concentrations ($d_p > 4$ nm) that spanned three orders of magnitude ($10^1 \sim 10^4$ cm$^{-3}$). The low background concentrations of aerosol particles in this region are driven by strong scavenging and relatively weak transport from lower latitudes, and those conditions can set the stage for the formation of UFP within the Arctic marine boundary layer, as shown by this and other studies. By combining two seasons of observations aboard CCGS *Amundsen* over a similar cruise track, this study was able to document conditions which led to sparse observations of UFP formation and growth in 2014 (events on 6% of measurement days) and more frequent observation of UFP formation and growth events in 2016 (events on 41% of measurement days). CS in the Canadian Arctic was consistently low (~1-2 h$^{-1}$) in both summer expeditions compared with other regions. Comparison of environmental conditions between the two seasons of measurements concludes that the generally low CS, coupled with higher solar radiation, a greater fraction of open water (lower sea ice concentration), and the differences in biological activity in the local marine environment collectively led to a greater frequency of UFP formation and growth in 2016. Given the general understanding of how meteorological conditions can influence UFP formation and growth (e.g., Engvall et al., 2008a; Heintzenberg et al., 2015; Kulmala et al., 2014), geographic similarities between UFP formation and growth events across the two expeditions stress the potential importance of marine microbial processes on the occurrence and behaviour of such events in the Canadian Arctic. Consequently, atmospheric chemistry of UFP formation and growth occurring in the Arctic may also be a key motivation for fundamental studies of biologically modified ocean-atmosphere interactions (Brüggemann et al., 2017; Lee et al., 2015; Prather et al., 2013).

## 5    Data Availability

NETCARE (Network on Climate and Aerosols: Addressing Key Uncertainties in Remote Canadian Environments, http://www.netcare-project.ca), which organized the field campaigns described in this work, is moving toward a publically available, online data archive. In the meantime, data can be accessed by contacting the principal investigator of the network: Jonathan Abbatt, University of Toronto (jabbatt@chem.utoronto.ca).



## 6    Acknowledgements

This research was conducted by the Network on Climate and Aerosols: Addressing Key Uncertainties in Remote Canadian Environments (NETCARE), funded by the Climate Change and Atmospheric Research (CCAR) program within the Natural Sciences and Engineering Research Council of Canada (NSERC). Funding to authors was also attributed to the National Centers of Excellence (NCE) ArcticNet and NSERC DG program. We thank Dr. Alex Lee and Dr. Luis Ladino for their help during ship mobilization in 2014, Prof. Jennifer Murphy for her assistance during the 2014 expedition, Dr. Alexander Moravek for his assistance during the 2016 expedition, Keith Levesque for overall coordination of both ship campaigns on behalf of ArcticNet, Claude Belzile and Mélanie Simard for DOC analysis, and the Canadian Coast Guard officers and crew of CCGS *Amundsen* for their tireless efforts to make these campaigns possible. We also thank Dr. Richard Leaitch for valuable discussions on the subject of this manuscript.

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

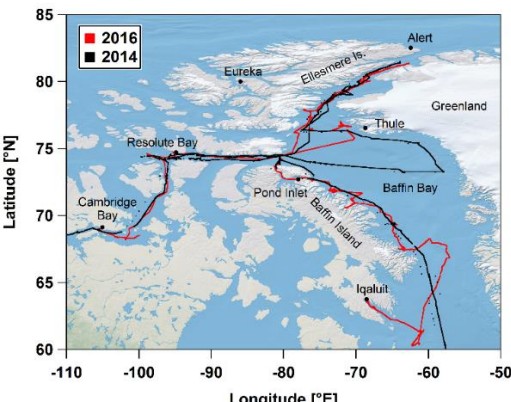

**Figure 1: Cruise track of CCGS *Amundsen* comprising the NETCARE 2014 and 2016 campaigns. In both expeditions, the ship progressed generally from the southern Baffin Bay, north toward Alert, and then south toward Resolute Bay and Cambridge Bay, traversing the Northwest Passage.**





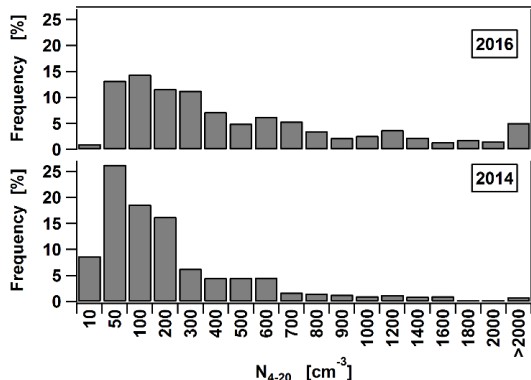

**Figure 2: Frequency distributions of ultrafine particle concentrations ($N_{4-20}$) based on 5-minute averaged measurements for the 2016 (top) and 2014 (bottom) expeditions. Bins are labelled with the upper limit, except for the rightmost bin, which is labelled with its lower limit (> 2000 cm$^{-3}$).**

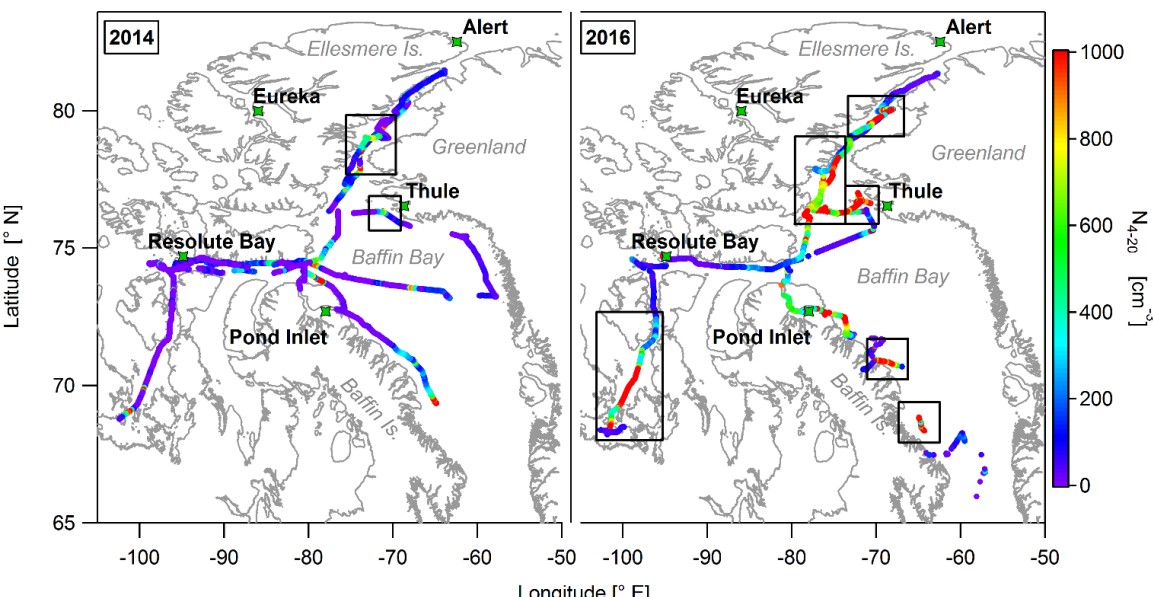

**Figure 3: Spatially resolved measurements of $N_{4-20}$ for the NETCARE 2014 (left) and 2016 (right) campaigns. Boxes denote locations where UFP formation and growth was observed.**





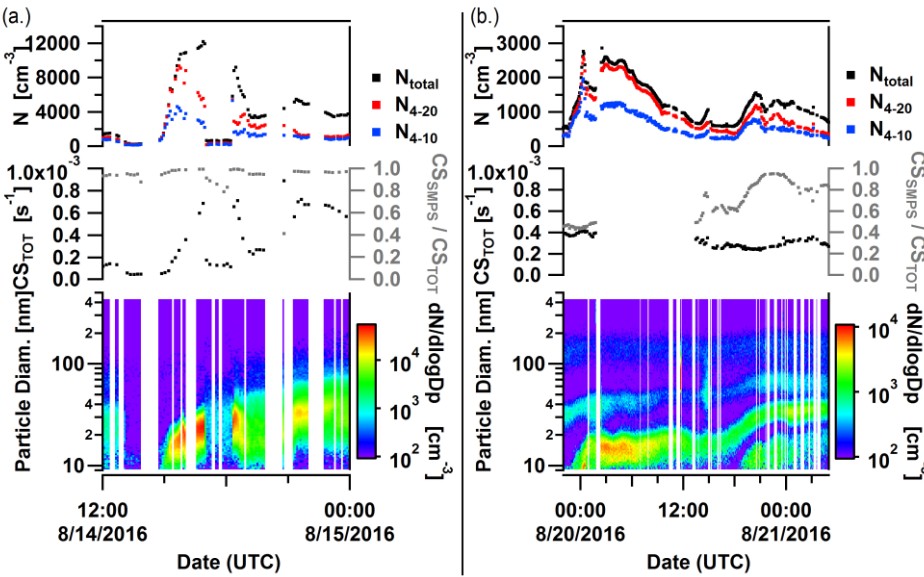

**Figure 4: Examples of a new particle formation and growth events observed during the NETCARE 2016 campaign. (a.) An example observed in the Nares Strait, during which the ship moved through a shallow fog, resulting in a period of low particle concentrations amidst a larger scale nucleation event. (b.) In the Peele Sound/Queen Maude Gulf region (south of Resolute Bay) nucleation was observed, and a subsequent nucleation event caused simultaneous growth of particles formed previously. Indeed, there is evidence for three modes growing simultaneously near the end of the event period. The top panels show 1-minute average particle concentrations in three size classes, $N_{total}$ ($d_p > 4$ nm), along with $N_{4-20}$ and $N_{4-10}$. Middle panels show the condensation sink ($CS_{TOT}$) and the fraction of CS contributed by particles with $d_p = 10-500$ nm ($CS_{SMPS} / CS_{TOT}$). The bottom panels are time-resolved SMPS size distributions plotted with the concentrations on a logarithmic scale. Note the difference in concentration scale between (a) and (b) in both top and bottom panels.**

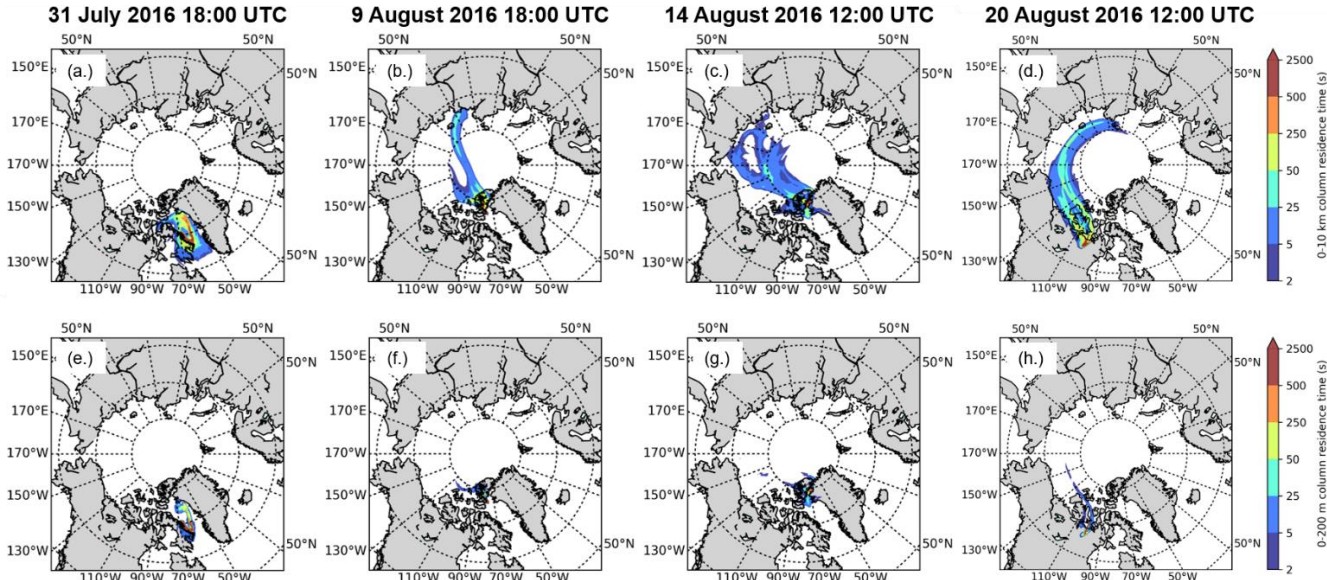





**Figure 5: FLEXPART air mass histories extending backwards 5 days from the time noted at the top of each pair of plots. The top row (a-d) represent time-integrated PES for a 0-10 km column, and the bottom row (e-h) represent time-integrated PES for a 0-200 m column. Each of the times corresponds to the observation of a UFP formation and/or growth event aboard CCGS *Amundsen*. Tracer release locations were dictated by the ship's coordinates at the given release time. Details on the model can be found in Section 2.2.**

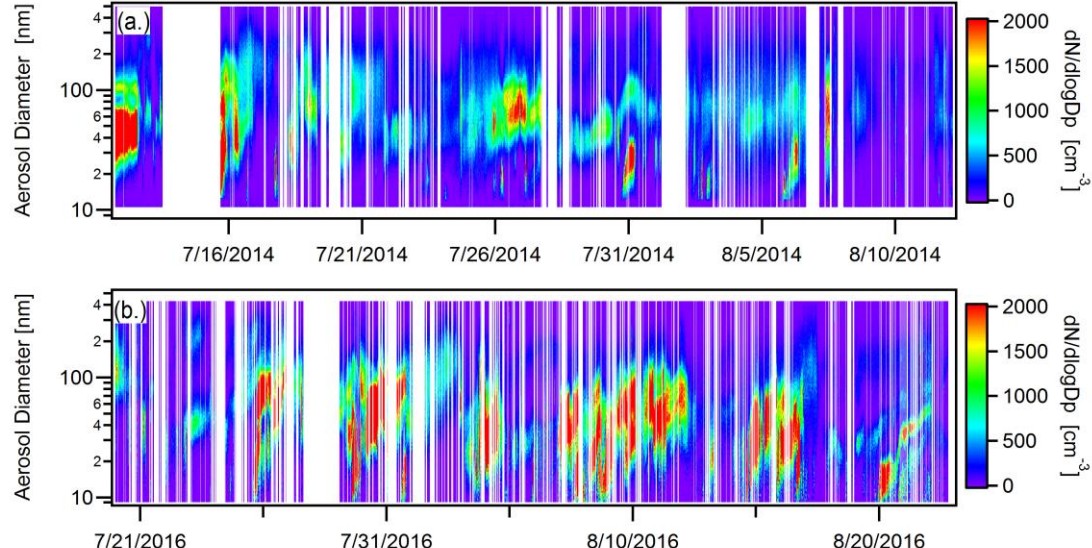

**Figure 6: Time series plots of SMPS data from 2014 (top) and 2016 (bottom) NETCARE campaigns on CCGS *Amundsen*. The prevalence of ultrafine particle formation and growth events in 2016 is generally visible throughout the campaign and is in stark comparison with the more isolated nucleation and growth events observed in 2014.**

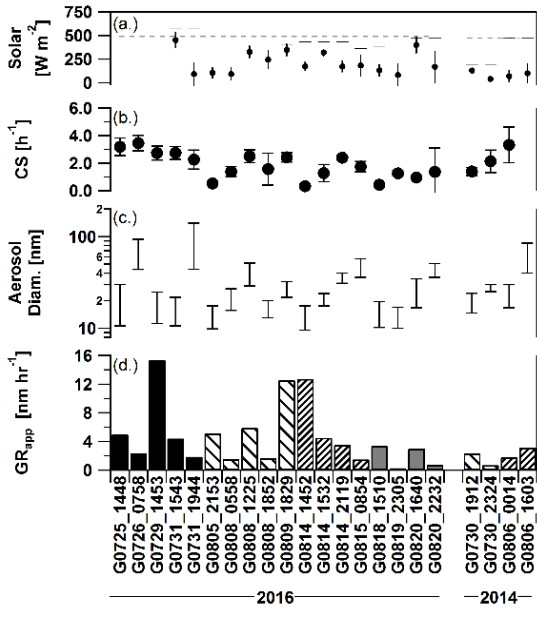



**Figure 7: Summary of parameters commonly relevant to ultrafine particle formation and growth in the atmosphere. In the top panel, filled circles represent the mean insolation (±1σ) for the period over which growth was measured, the horizontal dash marker is the maximum insolation for day on which the event began, and the horizontal grey dashed line is the average maximum incident shortwave radiation for days on which an event was *not* observed during each expedition. The second panel shows the condensation sink (CS; ±1σ) during each period in which growth rate was measured. The third panel shows the particle size range over which growth was observed/quantified. The bottom panel shows the apparent growth rate of particles (GR$_{app}$), measured as the rate of change in the modal diameter. The date and UTC time of the beginning of each event is coded within each event label, and the events are shaded to group by region.**

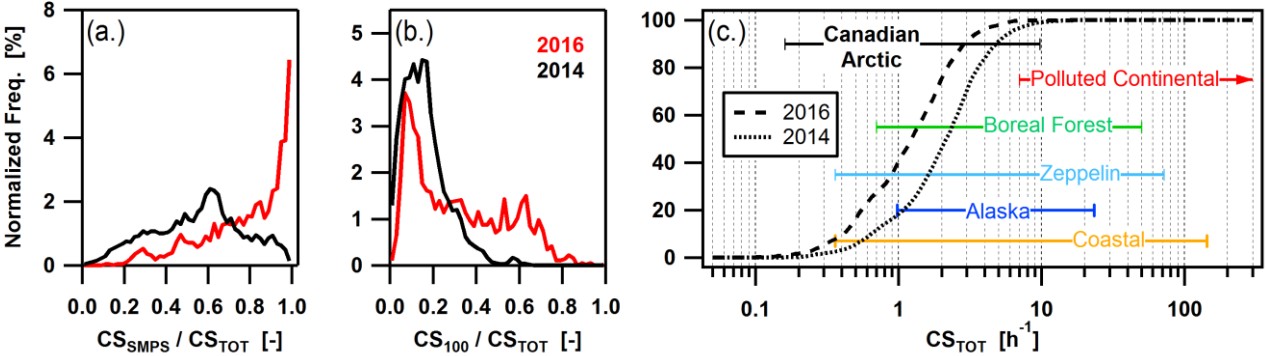

**Figure 8: (a.) Normalized frequency distribution of the fractional contribution of particles measured by SMPS (CS$_{SMPS}$; $d_p$ = 10 – 430 nm) to the total CS (CS$_{TOT}$) during the 2016 campaign. (b.) Normalized frequency distribution of the fractional contribution of particles with $d_p$ < 100 nm (CS$_{100}$) to CS$_{TOT}$. (c.) Normalized cumulative frequency distribution of CS calculated from measured size distributions during the 2014 and 2016 campaigns. Coloured horizontal lines, shown for context, denote ranges of CS from the Canadian Arctic (this study; 1% and 99% values), a polluted continental site (Po Valley, Italy), a boreal forest site (Hyytiälä, Finland)** (Westervelt et al., 2013)**, a site on the northern coast of Alaska (Utqiaġvik [Barrow], summer)** (Kolesar et al., 2017)**, Zeppelin station (near Ny Ålesund, Svalbard)** (Giamarelou et al., 2016)**, and a coastal location in the North Atlantic (Mace Head, Ireland)** (Dal Maso et al., 2002)**. The upper limit of the polluted continental site is off scale (~2×10$^4$ h$^{-1}$).**

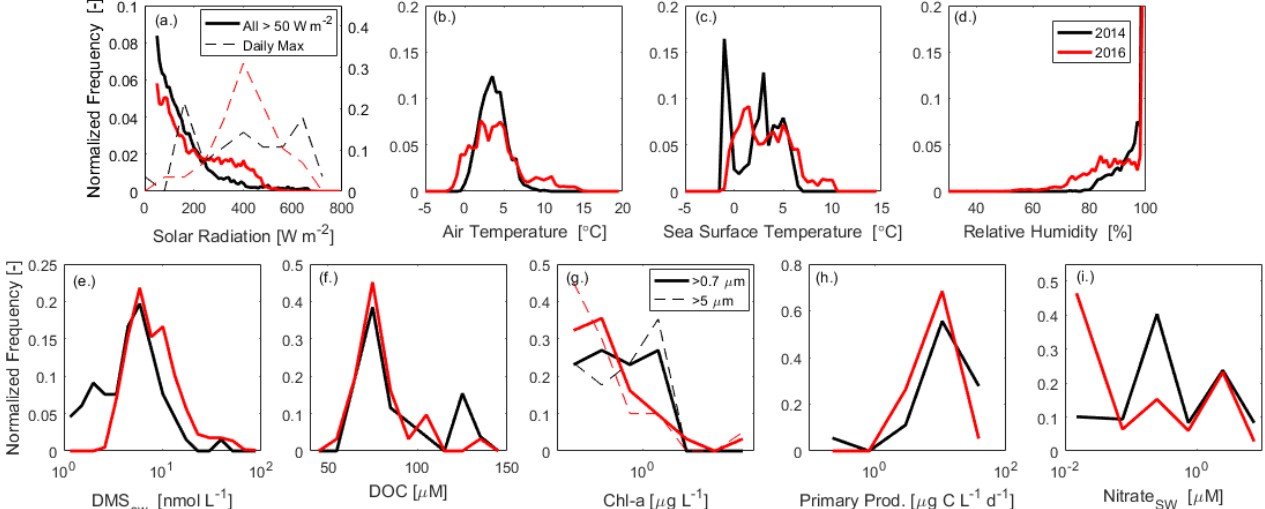

**Figure 9: Normalized histograms of environmental parameters measured or sampled aboard CCGS *Amundsen* in 2014 and 2016. (a.) Solar radiation, (b.) ambient air temperature, (c.) sea surface temperature, (d.) atmospheric relative humidity, (e.) dimethyl sulphide in surface seawater (DMS$_{SW}$), (f.) dissolved organic carbon (DOC) in surface seawater, (g.) chlorophyll-a (chl-a) in surface**





seawater captured on filters of two different porosities, (h.) primary productivity in surface seawater, and (i.) nitrate in seawater averaged over the top 35 meters of each depth profile.

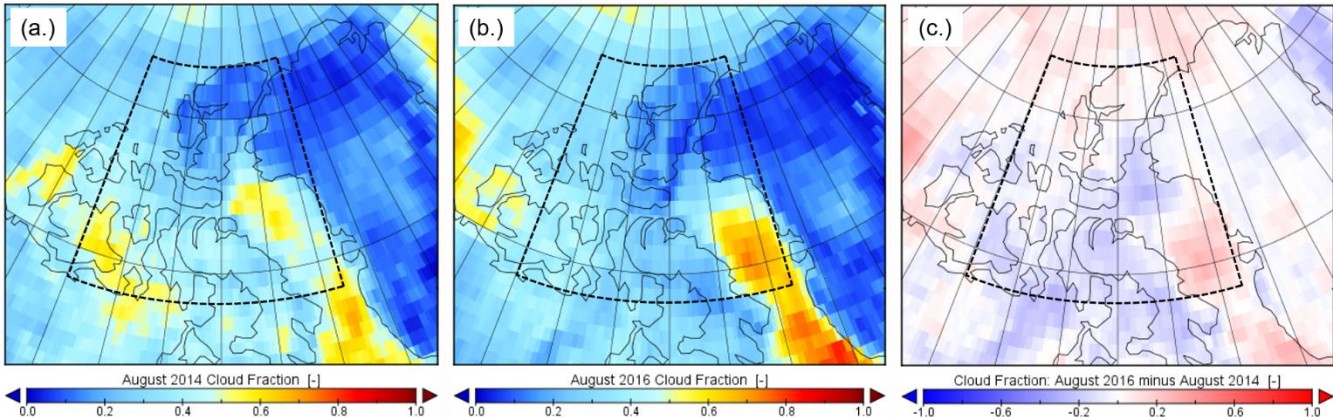

**Figure 10: MODIS-Aqua retrieval of monthly-averaged daily liquid cloud fraction for (a.) August 2014, (b.) August 2016, and (c.) the difference between 2016 and 2014. The dashed black box denotes the area within which the study-area average was computed. Data from NASA/GSFC on a Lambert Conformal Conic projection.**

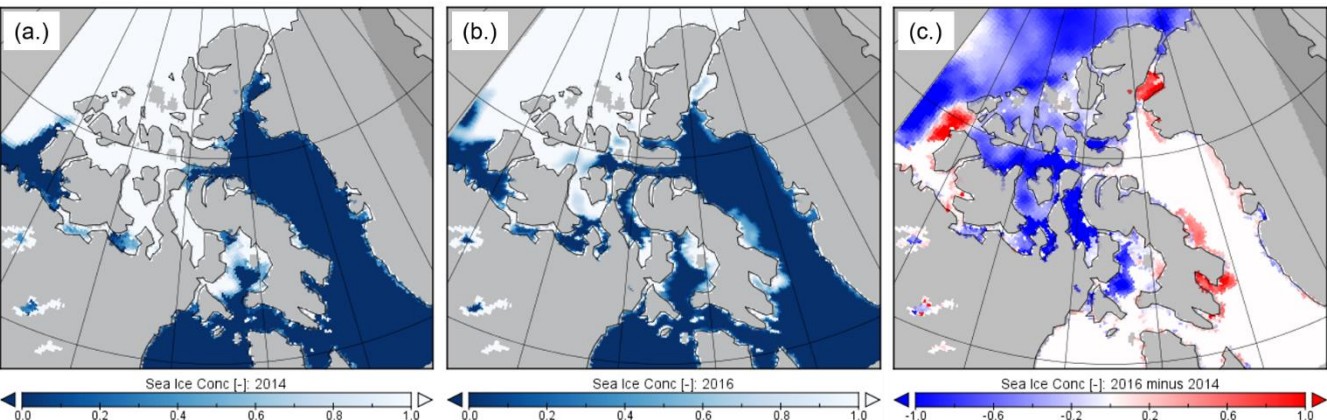

**Figure 11: Sea ice concentrations within the Canadian Arctic, plotted for a single example day (1 August): (a.) 2014, (b.) 2016, and (c.) difference between 2016 and 2014. Data from NASA/NSIDC on a Lambert Conformal Conic projection.**