# Peer review of "Frequent Ultrafine Particle Formation and Growth in the Canadian Arctic Marine Environment"

_Atmospheric Chemistry and Physics, 2017_

## Referee Comment (RC1) · Anonymous Referee #1 · 23 Jun 2017

This manuscript summarizes measurements conducted in 2014 and 2016 in the Canadian Arctic aboard the CCGS Amundsen. Ultrafine particle formation and growth were frequently encountered especially in 2016. In addition to summarizing the particle events the authors look at various meteorological and oceanographic data that might explain the differences in particle events between 2014 and 2016. Unfortunately the differences in atmospheric and oceanic conditions were small and there was no smoking gun that clearly explains the differences in particle events between the two years.

The manuscript is well written and easy to read. I feel the manuscript is appropriate for ACP and should be published with minor revisions.

[Figure]

1. I think it should be clearly pointed out that these are coastal measurements and not the open Arctic Ocean. I think "Coastal" should be added to the title before Canadian.

2. Page 10, line 15. What VOCs?

3. Figure 5, e-h. The Flexpart PES for 0-200m are very hard to read. Can these figures be expanded to better show the data? This really points to the local source of the particles.

4. Page 17 line 15. Saying these factors "led to a greater frequency of UFP formation" is really speculation. Perhaps "could have individually or collectively contributed to a greater frequency..."

---

## Referee Comment (RC2) · Anonymous Referee #2 · 4 Aug 2017

Frequent Ultrafine Particle Formation and Growth in the Canadian Arctic Marine Environment by Collins et al., 2017 ACPD

Atmos. Chem. Phys. Discuss., https://doi.org/10.5194/acp-2017-411 Manuscript under review for journal Atmos. Chem. Phys.

Collins et al. (2017) is a great piece of work characterising aerosol in a difficult environment (the Arctic), on a challenging platform (an icebreaker) for two different years (2014 and 2016). I congratulate very much to the authors, and to the Canadian program NETCARE which should be an example of interdisciplinary studies to follow.

I think the paper should be well accepted in ACP, following few major revisions which I

am confident the author will be able to carry out.

1) It is mentioned a number of times in the text that the Arctic marine microbial communities are likely to be responsible for the large increase of new particle formation recorded in 2016 vs 2014 (41% and 6% of the time, respectively). For example, on page 15 line 16-18 "The source strengths of gas-phase precursors and reactive species, which are generally not well understood in the marine environment, are key remaining factors for explaining differences in ultrafine particle production". Reading section 3.3, one may get the feeling that this is the main reason for the large increase observed in 2016 relative to 2014, given other meteorological and physical conditions did not change substantially. I suggest to modify the abstract (a bit too general in the current state) and report - for example - important conclusion such as line 33 pg 13 "CS may not be a factor that directly limits the formation of UPF in this region". I think is important to stress that chemical precursors (likely coming from Arctic marine communities) may play an important rule in increasing UPF, and physical conditions (different CS, for example) may not be as important as chemical precursors availability. If that is the case, it should be stressed in the abstract, in the current stage too general and not representative of the discussions and conclusions presented across the manuscript.

2) Figure S5 should be a main part of the paper (maybe as new Figure 12) because it stresses a major difference across the two different years (differences up to 13-25% in sea ice concentrations) - therefore associating UFP events to open water, higher percentages of sea ice marginal zones, and less packed ice. On this regards, the authors should refer to a recent paper (Arctic sea ice melt leads to new particle formation, Dall'Osto et al., 2017a, Scientific Reports | 7: 3318 | DOI:10.1038/s41598-017-03328-1), where air mass trajectory analysis and atmospheric nitrogen and sulphur tracers linked frequent nucleation events to biogenic precursors released by open water and melting sea ice Arctic regions. Additionally, when discussing this (I leave to the author if prefer to discuss this in the result discussion part or in the conclusion paper) they should also discuss this potential source (polar open water and sea ice marginal Arctic

sea ice regions) and put into context another recent paper from NETCARE (Croft et al, 2016, Nature Comm, another possible source of UFP related to bird colonies).

Minor comments

- Section 3.3.2 - oceanic conditions. I congratulate to the authors for improving the paper with this interdisciplinary part, an interesting one. Whilst DOC - among other marine biological measurements - during different years were almost identical, Figure S6 shows interesting differences for nitrate (among other variables), suggesting phytoplankton production season was more advanced and well into post bloom phase during 2016 (relative to 2016). A recent paper (Dall'Osto et al., 2017, Antarctic sea ice is a source of organic nitrogen in aerosols, Scientific Report, DOI:10.1038/s41598-017-06188-x) also go in the same direction, showing sea ice marginal region with more advanced post bloom phase enhanced in UFP. It may be worth to stress that in polar regions (both Antarctic and Arctic) the biology is playing a role (and seems not Chl, but the stage of the bloom, is the key factor) and more interdisciplinary studies are needed.

- pg 3 line 30-25. Whilst the authors do a decent job in addressing the different chemical precursors, it may be more appropriate to cite only works carried out in Arctic regions (not Atlantic or other oceans) and not forget Sippila et al 2016 (Nature) and also to address recent new findings (Croft et al., 2016, Birds colony emissions) and marginal sea ice (Dall'Osto et al., 2017a, Scien Rep).

- Pg 11 line 2-5, it is possible to access the importance of coastal vs open ocean sources? As Rev 1 suggests, is this study more related to a specific environment, such as Archipelagos, and not to be extrapolated to open ocean and marginal sea ice zone Arctic areas?

- pg 15 line 23 - I think it is figure S5

-pg 15 line 22. I think the authors should improve this section and decide what is more appropriate (if include figure S5 as figure 12, and expand this section). I think maybe

presenting an average map for the two different seasons (2014 and 2016) but I am not sure the 1st of August is representative, I would use a longer period, or present figure S5 in the main text.

---

## Author Comment (AC2) · 13 Sep 2017

*Response to Review*

**Frequent Ultrafine Particle Formation and Growth in Canadian Arctic Marine and Coastal Environments**

Douglas B. Collins[1], Julia Burkart[1], Rachel Y.-W. Chang[2], Martine Lizotte[3], Aude Boivin-Rioux[4], Marjolaine Blais[4], Emma L. Mungall[1], Matthew Boyer[2], Victoria E. Irish[5], Guillaume Massé[3], Daniel Kunkel[6], Jean-Éric Tremblay[3], Tim Papakyriakou[7], Allan K. Bertram[5], Heiko Bozem[6], Michel Gosselin[4], Maurice Levasseur[3], Jonathan P.D. Abbatt[1]

[1]Department of Chemistry, University of Toronto, Toronto, ON, M5S 3H6, Canada
[2]Department of Physics and Atmospheric Science, Dalhousie University, Halifax, NS, B3H 4R2, Canada
[3]Département de biologie (Québec-Océan), Université Laval, Québec, QC, G1V 0A6, Canada
[4]Institut des sciences de la mer de Rimouski, Université du Québec à Rimouski, Rimouski, QC, G5L 3A1, Canada
[5]Department of Chemistry, University of British Columbia, Vancouver, BC, V6T 1Z1, Canada
[6]Johannes Gutenberg University of Mainz, Institute of Atmospheric Physics, 55099 Mainz, Germany
[7]Center for Earth Observation Science, University of Manitoba, Winnipeg, MB, R3T 2N2, Canada

*Correspondence to*: Douglas B. Collins (douglas.collins@utoronto.ca)

Referee comments are reproduced in blue. Author responses are provided in black.

**Anonymous Referee #2:**

Collins et al. (2017) is a great piece of work characterising aerosol in a difficult environment (the Arctic), on a challenging platform (an icebreaker) for two different years (2014 and 2016). I congratulate very much to the authors, and to the Canadian program NETCARE which should be an example of interdisciplinary studies to follow. I think the paper should be well accepted in ACP, following few major revisions which I am confident the author will be able to carry out.

The authors thank the referee for their thoughtful review of the manuscript and appreciate their kind words about the NETCARE program.

1) It is mentioned a number of times in the text that the Arctic marine microbial communities are likely to be responsible for the large increase of new particle formation recorded in 2016 vs 2014 (41% and 6% of the time, respectively). For example, on page 15 line 16-18 "The source strengths of gas-phase precursors and reactive species, which are generally not well understood in the marine environment, are key remaining factors for explaining differences in ultrafine particle production". Reading section 3.3, one may get the feeling that this is the main reason for the large increase observed in 2016 relative to 2014, given other meteorological and physical conditions did not change substantially. I suggest to modify the abstract (a bit too general in the current state) and report - for example - important conclusion such as line 33 pg 13 "CS may not be a factor that directly limits the formation of UPF in this region". I think is important to stress that chemical precursors (likely coming from Arctic marine communities) may play an important rule in increasing UPF, and physical conditions (different CS, for example) may not be as important as chemical precursors availability. If that is the case, it should be stressed in the abstract, in the current stage too general and not representative of the discussions and conclusions presented across the manuscript.

Given the lack of a clear delineation of the importance of biogeochemical factors versus meteorological factors leading to the formation and growth of UFP within this study, the authors feel it is not prudent to change the abstract to reflect greater certainty in the conclusion of a biogenic driver behind inter-annual differences in UFP formation. Indeed, our findings are suggestive that the differences in UFP formation

and growth *may* be tied to differences in sea ice coverage and/or the phase of biological activity in the Canadian Arctic – and this study provides an interdisciplinary analysis of key observations. However, decreased cloud fraction and increased solar radiation were also documented in 2016 compared with 2014, and may also help lead to greater UFP formation and/or growth. Ongoing study within NETCARE using computational tools may help clarify conclusions about which factors may be driving UFP formation in this region. We have added a statement in the conclusions highlighting the importance of computational tools to deriving insight on the mechanisms behind UFP formation and growth.

Page 18, lines 18-20 – sentence added: "In addition, chemical transport models and other computational tools, in conjunction with the observations reported in this study, may lend important insight into the drivers of UFP formation in the Canadian Arctic, and may be translatable to other remote locations."

2) Figure S5 should be a main part of the paper (maybe as new Figure 12) because it stresses a major difference across the two different years (differences up to 13-25% in sea ice concentrations) - therefore associating UFP events to open water, higher percentages of sea ice marginal zones, and less packed ice. On this regards, the authors should refer to a recent paper (Arctic sea ice melt leads to new particle formation, Dall'Osto et al., 2017a, Scientific Reports | 7: 3318 | DOI:10.1038/s41598-017-03328-1), where air mass trajectory analysis and atmospheric nitrogen and sulphur tracers linked frequent nucleation events to biogenic precursors released by open water and melting sea ice Arctic regions. Additionally, when discussing this (I leave to the author if prefer to discuss this in the result discussion part or in the conclusion paper) they should also discuss this potential source (polar open water and sea ice marginal Arctic sea ice regions) and put into context another recent paper from NETCARE (Croft et al, 2016, Nature Comm, another possible source of UFP related to bird colonies).

The referee's suggestion to include Figure S5 in the main text (as Figure 12) has been included in the revisions. While a short discussion addressing the ability of discontinuous ice to enable air-sea interaction was present in the initial manuscript, references to the work by Dall'Osto et al. (2017b) and Sharma et al. (2012) have now been included, and a further concluding statement at the end of the paragraph (p16, lines 2-5) has been added. The authors agree with the referee that the change in sea ice could lead to differences in air-sea exchange of volatile UFP precursors, and this has been clarified in the revised manuscript. A statement (p16, lines 18-21) connecting the discussion to coastal sources of UFP precursors has also been added, including not only bird colonies (Croft et al., 2016; Weber et al., 1998), but also intertidal zones (O'Dowd et al., 2002; Sipilä et al., 2016). In addition, a discussion of the interaction of marginal sea ice and atmospheric chemistry in polar regions has now been added to the manuscript (p17, lines 21-29).

Minor comments

- Section 3.3.2 - oceanic conditions. I congratulate to the authors for improving the paper with this interdisciplinary part, an interesting one. Whilst DOC - among other marine biological measurements - during different years were almost identical, Figure S6 shows interesting differences for nitrate (among other variables), suggesting phytoplankton production season was more advanced and well into post bloom phase during 2016 (relative to 2016). A recent paper (Dall'Osto et al., 2017, Antarctic sea ice is a source of organic nitrogen in aerosols, Scientific Report, DOI:10.1038/s41598-017-06188-x) also go in the same direction, showing sea ice marginal region with more advanced post bloom phase enhanced in UFP. It may be worth to stress that in polar regions (both Antarctic and Arctic) the biology is playing a role (and seems not Chl, but the stage of the bloom, is the key factor) and more interdisciplinary studies are needed.

As the referee describes, the manuscript discusses the importance of the bloom phase on marine biosphere-atmosphere interactions (e.g., p16 starting on line 10). The conclusion that the bloom state was different between the two summers was reached by holistically analyzing the nitrate concentration near the sea

surface, primary production rates, dimethyl sulphide concentrations in seawater, and the nuanced information content within the size-segregated chlorophyll concentrations. The importance of post-bloom phase biogeochemistry on air-sea interactions has also been noted in various previous studies (Collins et al., 2013; Lee et al., 2015; O'Dowd et al., 2015; Wang et al., 2015), which are cited within the manuscript along with foundational studies of bloom dynamics (Azam et al., 1983). Indeed, biogeochemistry in the marginal ice zone is germane to this study as CCGS *Amundsen* encountered sea ice commonly during both summers, and sea ice concentrations in the region were a major distinguishing factor between the 2014 and 2016 expeditions. A discussion of the importance of the marginal ice zone to atmospheric chemistry in polar regions is warranted and has been added to the manuscript (p17, lines 21-29). It is important to point out that the impact of specific sources of UFP formation and growth precursors were not scrutinized in this study, as air mass histories indicate the inclusion of a variety of potential source characterizations (e.g., coastal, open water, sea ice) for each UFP formation event described in the manuscript. Consequently, a more broad approach to characterize the region as a whole was undertaken.

*- pg 3 line 30-25. Whilst the authors do a decent job in addressing the different chemical precursors, it may be more appropriate to cite only works carried out in Arctic regions (not Atlantic or other oceans) and not forget Sippila et al 2016 (Nature) and also to address recent new findings (Croft et al., 2016, Birds colony emissions) and marginal sea ice (Dall'Osto et al., 2017a, Scien Rep).*

Since so little is known about marine emissions of UFP precursors (and as the literature on aerosol nucleation chemistry develops in parallel), the authors feel that it is important to include relevant findings from various marine regions in light of the desire for process-level understanding in the long run. Relatively few measurements have been made to connect UFP formation with specific precursors, so it is important to keep an open mind on the possible chemical drivers. While it is becoming clear that organic material leads to particle growth in the Arctic (Willis et al., 2016), and that an accurate accounting for ammonia is important for modeling UFP formation (Croft et al., 2016), the level of understanding in the marine atmosphere lags behind the terrestrial biosphere to a great degree. The key points about sources of N-containing material in secondary marine aerosol and in biogenic material retrieved from sea ice are important contributors to accounting for possible marine UFP precursors (e.g., Dall'Osto et al., 2017a; Facchini et al., 2008). As noted by the referee, UFP formation in coastal and marginal ice zones have been observed but much more information is needed to understand these processes fully. Iodine chemistry that is characteristic of intertidal emissions (Allan et al., 2015; Sipilä et al., 2016) and marginal ice emissions of sulfur- and nitrogen-containing compounds (Dall'Osto et al., 2017a; Levasseur, 2013) are of interest to the community, and are now appropriately acknowledged and cited within the Introduction. (p3, lines 30-34)

*- Pg 11 line 2-5, it is possible to access the importance of coastal vs open ocean sources? As Rev 1 suggests, is this study more related to a specific environment, such as Archipelagos, and not to be extrapolated to open ocean and marginal sea ice zone Arctic areas?*

As noted in a prior comment to this referee (*vide supra*), the comparative analysis of marine biological parameters and atmospheric aerosol observations are deliberately broad in nature. Since the Canadian Arctic can have influences from coastal margins, open water regions, and transport from the high Arctic, the authors felt it prudent to retain a regional scale analysis that included all of these sources in a 'synoptic' sense. At the suggestion of the referees, a more developed discussion of the impacts of marginal ice, coastal seabirds, and intertidal zones has been included in various parts of Section 3 to clarify for the reader the potential importance of such sources to UFP formation and growth in this region. More targeted study of each of these sources may be required to understand their impacts on the environment as a whole.

This figure reference has been changed to 'Figure 12' as Figure S5 has been moved to the main text.

-pg 15 line 22. I think the authors should improve this section and decide what is more appropriate (if include figure S5 as figure 12, and expand this section). I think maybe presenting an average map for the two different seasons (2014 and 2016) but I am not sure the 1st of August is representative, I would use a longer period, or present figure S5 in the main text.

In accordance with another of this referee's comments, Figure S5 has been incorporated in the main text as Figure 12, which the authors feel enhances the manuscript in an important way. The spatial representation of Figure 11 remains unchanged in the revised manuscript, as it is provided as an example of the pattern of differences in sea ice concentration throughout the region, but the quantitative changes are well represented by Figure 12. In addition, the connection between differences in sea ice concentration and biological activity have been further developed in the context of UFP formation throughout Section 3.3.2, in response to the referee's comments. Discussions of interactions between aerosol chemistry and transport over open ocean (Sharma et al., 2012), associations between sea ice concentration and UFP formation (Dall'Osto et al., 2017b), and possible influences of sea ice on marine biology in the surface ocean (Dall'Osto et al., 2017a; Levasseur, 2013) have been enhanced or added to the manuscript.

**References**

Allan, J. D., Williams, P. I., Najera, J., Whitehead, J. D., Flynn, M. J., Taylor, J. W., Liu, D., Darbyshire, E., Carpenter, L. J., Chance, R., Andrews, S. J., Hackenberg, S. C. and McFiggans, G.: Iodine observed in new particle formation events in the Arctic atmosphere during ACCACIA, Atmos. Chem. Phys., 15(10), 5599–5609, doi:10.5194/acp-15-5599-2015, 2015.

Azam, F., Field, J. G., Graf, J. S., Meyer-Rei, L. A. and Thingstad, F.: The Ecological Role of Water-Column Microbes in the Sea, Mar. Ecol. Prog. Ser., 10, 257–263, 1983.

Collins, D. B., Ault, A. P., Moffet, R. C., Ruppel, M. J., Cuadra-Rodriguez, L. A., Guasco, T. L., Corrigan, C. E., Pedler, B. E., Azam, F., Aluwihare, L. I., Bertram, T. H., Roberts, G. C., Grassian, V. H. and Prather, K. A.: Impact of marine biogeochemistry on the chemical mixing state and cloud forming ability of nascent sea spray aerosol, J. Geophys. Res. Atmos., 118(15), 8553–8565, doi:10.1002/jgrd.50598, 2013.

Croft, B., Wentworth, G. R., Martin, R. V., Leaitch, W. R., Murphy, J. G., Murphy, B. N., Kodros, J. K., Abbatt, J. P. D. and Pierce, J. R.: Contribution of Arctic seabird-colony ammonia to atmospheric particles and cloud-albedo radiative effect, Nat. Commun., 7, 13444, doi:10.1038/ncomms13444, 2016.

Dall'Osto, M., Ovadnevaite, J., Paglione, M., Beddows, D. C. S., Ceburnis, D., Cree, C., Cortés, P., Zamanillo, M., Nunes, S. O., Pérez, G. L., Ortega-Retuerta, E., Emelianov, M., Vaqué, D., Marrasé, C., Estrada, M., Montserrat Sala, M., Vidal, M., Fitzsimons, M. F., Beale, R., Airs, R., Rinaldi, M., Decesari, S., Facchini, M. C., Harrison, R. M., O'Dowd, C. and Simó, R.: Antarctic sea ice region as a source of biogenic organic nitrogen in aerosols, Sci. Rep., 7, 6047, doi:10.1038/s41598-017-06188-x, 2017a.

Dall'Osto, M., Beddows, D. C. S., Tunved, P., Krejci, R., Ström, J., Hansson, H.-C., Yoon, Y. J., Park, K.-T., Becagli, S., Udisti, R., Onasch, T., O´Dowd, C. D., Simó, R. and Harrison, R. M.: Arctic sea ice melt leads to atmospheric new particle formation, Sci. Rep., 7, 3318, doi:10.1038/s41598-017-03328-1, 2017b.

Facchini, M. C., Decesari, S., Rinaldi, M., Carbone, C., Finessi, E., Mircea, M., Fuzzi, S., Moretti, F., Tagliavini, E., Ceburnis, D. and O'Dowd, C. D.: Important Source of Marine Secondary Organic Aerosol from Biogenic Amines, Environ. Sci. Technol., 42(24), 9116–9121, doi:10.1021/es8018385, 2008.

Lee, C., Sultana, C. M., Collins, D. B., Santander, M. V., Axson, J. L., Malfatti, F., Cornwell, G. C., Grandquist, J. R., Deane, G. B., Stokes, M. D., Azam, F., Grassian, V. H. and Prather, K. A.: Advancing Model Systems for Fundamental Laboratory Studies of Sea Spray Aerosol Using the Microbial Loop, J. Phys. Chem. A, 119(33), 8860–8870, doi:10.1021/acs.jpca.5b03488, 2015.

Levasseur, M.: Impact of Arctic meltdown on the microbial cycling of sulphur, Nat. Geosci., 6(9), 691–700, doi:10.1038/ngeo1910, 2013.

O'Dowd, C., Ceburnis, D., Ovadnevaite, J., Bialek, J., Stengel, D. B., Zacharias, M., Nitschke, U., Connan, S., Rinaldi, M., Fuzzi, S., Decesari, S., Facchini, M. C., Marullo, S., Santoleri, R., Dell'Anno, A., Corinaldesi, C., Tangherlini, M. and Danovaro, R.: Connecting marine productivity to sea-spray via nanoscale biological processes: Phytoplankton Dance or Death Disco?, Sci. Rep., 5, 14883, doi:10.1038/srep14883, 2015.

O'Dowd, C. D., Jimenez, J. L., Bahreini, R., Flagan, R. C., Seinfeld, J. H., Hämeri, K., Pirjola, L., Kulmala, M., Jennings, S. G. and Hoffmann, T.: Marine aerosol formation from biogenic iodine emissions, Nature, 417(6889), 632–636, doi:10.1038/nature00775, 2002.

Sharma, S., Chan, E., Ishizawa, M., Toom-Sauntry, D., Gong, S. L., Li, S. M., Tarasick, D. W., Leaitch, W. R., Norman, A., Quinn, P. K., Bates, T. S., Levasseur, M., Barrie, L. A. and Maenhaut, W.: Influence of transport and ocean ice extent on biogenic aerosol sulfur in the Arctic atmosphere, J. Geophys. Res. Atmos., 117(D12), D12209, doi:10.1029/2011JD017074, 2012.

Sipilä, M., Sarnela, N., Jokinen, T., Henschel, H., Junninen, H., Kontkanen, J., Richters, S., Kangasluoma, J., Franchin, A., Peräkylä, O., Rissanen, M. P., Ehn, M., Vehkamäki, H., Kurten, T., Berndt, T., Petäjä, T., Worsnop, D., Ceburnis, D., Kerminen, V.-M., Kulmala, M. and O'Dowd, C.: Molecular-scale evidence of aerosol particle formation via sequential addition of HIO3, Nature, 537, 532–534, doi:10.1038/nature19314, 2016.

Wang, X., Sultana, C. M., Trueblood, J., Hill, T. C. J., Malfatti, F., Lee, C., Laskina, O., Moore, K. A., Beall, C. M., McCluskey, C. S., Cornwell, G. C., Zhou, Y., Cox, J. L., Pendergraft, M. A., Santander, M. V., Bertram, T. H., Cappa, C. D., Azam, F., DeMott, P. J., Grassian, V. H. and Prather, K. A.: Microbial Control of Sea Spray Aerosol Composition: A Tale of Two Blooms, ACS Cent. Sci., 1(3), 124–131, doi:10.1021/acscentsci.5b00148, 2015.

Weber, R. J., McMurry, P. H., Mauldin, L., Tanner, D. J., Eisele, F. L., Brechtel, F. J., Kreidenweis, S. M., Kok, G. L., Schillawski, R. D. and Baumgardner, D.: A study of new particle formation and growth involving biogenic and trace gas species measured during ACE 1, J. Geophys. Res. Atmos., 103(D13), 16385–16396, doi:10.1029/97JD02465, 1998.

Willis, M. D., Burkart, J., Thomas, J. L., Köllner, F., Schneider, J., Bozem, H., Hoor, P. M., Aliabadi, A. A., Schulz, H., Herber, A. B., Leaitch, W. R. and Abbatt, J. P. D.: Growth of nucleation mode particles in the summertime Arctic: a case study, Atmos. Chem. Phys., 16(12), 7663–7679, doi:10.5194/acp-16-7663-2016, 2016.

---

## Author Response (AR1)

*Response to Review*

**Frequent Ultrafine Particle Formation and Growth in Canadian Arctic Marine and Coastal Environments**

Douglas B. Collins[1], Julia Burkart[1], Rachel Y.-W. Chang[2], Martine Lizotte[3], Aude Boivin-Rioux[4], Marjolaine Blais[4], Emma L. Mungall[1], Matthew Boyer[2], Victoria E. Irish[5], Guillaume Massé[3], Daniel Kunkel[6], Jean-Éric Tremblay[3], Tim Papakyriakou[7], Allan K. Bertram[5], Heiko Bozem[6], Michel Gosselin[4], Maurice Levasseur[3], Jonathan P.D. Abbatt[1]

[1]Department of Chemistry, University of Toronto, Toronto, ON, M5S 3H6, Canada
[2]Department of Physics and Atmospheric Science, Dalhousie University, Halifax, NS, B3H 4R2, Canada
[3]Québec-Océan, Département de biologie, Université Laval, Québec, QC, G1V 0A6, Canada
[4]Institut des sciences de la mer de Rimouski, Université du Québec à Rimouski, Rimouski, QC, G5L 3A1, Canada
[5]Department of Chemistry, University of British Columbia, Vancouver, BC, V6T 1Z1, Canada
[6]Johannes Gutenberg University of Mainz, Institute of Atmospheric Physics, 55099 Mainz, Germany
[7]Center for Earth Observation Science, University of Manitoba, Winnipeg, MB, R3T 2N2, Canada

*Correspondence to*: Douglas B. Collins (douglas.collins@utoronto.ca)

Referee comments are reproduced in blue. Author responses are provided in black.

**Anonymous Referee #1**:

This manuscript summarizes measurements conducted in 2014 and 2016 in the Canadian Arctic aboard the CCGS Amundsen. Ultrafine particle formation and growth were frequently encountered especially in 2016. In addition to summarizing the particle events the authors look at various meteorological and oceanographic data that might explain the differences in particle events between 2014 and 2016. Unfortunately the differences in atmospheric and oceanic conditions were small and there was no smoking gun that clearly explains the differences in particle events between the two years. The manuscript is well written and easy to read. I feel the manuscript is appropriate for ACP and should be published with minor revisions.

The authors appreciate the assessment and accurate summary of the main points of the manuscript.

1. I think it should be clearly pointed out that these are coastal measurements and not the open Arctic Ocean. I think "Coastal" should be added to the title before Canadian.

The authors agree with the reviewer's point that the environment sampled aboard CCGS *Amundsen* during both expeditions is often proximal to a coastline. While air mass histories that consider the 0-10 km column height show appreciable residence over the central Arctic Ocean, the spatial component of near surface air mass history is restricted to a small region (c.f. Item 3 from Referee #1) which is often overlapping a coastline. Indeed, coastal impacts are of possible importance to ultrafine particle (UFP) formation and growth phenomena in this region, as discussed in this manuscript (e.g., p10 starting at line 24) and the relevant references cited therein. The authors feel that the study is both coastal and marine, so both terms now appear in the revised title:

"Frequent Ultrafine Particle Formation and Growth in Canadian Arctic Marine and Coastal Environments"

2. Page 10, line 15. What VOCs?

The reference to Mungall et al. (2017) is directed at recent measurements of oxygenated volatile organic compounds (VOCs) conducted on the CCGS *Amundsen* expedition in 2014 using acetate chemical ionization mass spectrometry. Factor analysis of the data revealed a unique contribution of an ocean source

to formic acid, isocyanic acid, and organic oxo-acids of varying carbon number. A larger array of compounds may have been present and co-emitted with compounds attributed to the 'Ocean Factor' in that study but do not have a greater proton affinity than acetate, and so cannot be measured by that technique. For this reason, it is prudent to retain a general characterization of the possible compounds, as little is known about the profile of VOCs emitted from the ocean and their dependence on biological and/or chemical interactions at the air-sea interface.

The manuscript now specifies the VOCs as oxygenated: (p10, line 14-15) "…and evidence of an ocean source for oxygenated volatile organic compounds…".

3. Figure 5, e-h. The Flexpart PES for 0-200m are very hard to read. Can these figures be expanded to better show the data? This really points to the local source of the particles.

The referee's comment reflects a conceptual decision on how the figure was designed. The 8-panel figure was intended to illustrate the history of relevant air masses over the Arctic Ocean in addition to the finding that the relevant air masses were in contact with the boundary layer/Earth's surface only within a limited spatial extent around the model 'release' point (ship location). The referee's recommendation to expand the spatial scale of Figure 5(e-h) has been incorporated in the revised manuscript with a specific note added to the caption highlighting the vastly different spatial scales used in the top versus the bottom row of panels. In order to achieve a readable map with a finer spatial scale, a slightly different projection was used for the new maps.

4. Page 17 line 15. Saying these factors "led to a greater frequency of UFP formation" is really speculation. Perhaps "could have individually or collectively contributed to a greater frequency..."

The measured language suggested by the referee has been adopted in the revised manuscript. The excerpt on p17 line 15-16 is now: "…and the differences in the biological activity in the local marine environment may have individually or collectively contributed to a greater frequency in UFP formation and growth in 2016." Indeed this study was not able to unequivocally identify a driving force for UFP formation and growth, but we hope that future chemical transport model studies that are constrained by the data reported here may help elucidate factors that lead to the observations.

**Anonymous Referee #2:**

Collins et al. (2017) is a great piece of work characterising aerosol in a difficult environment (the Arctic), on a challenging platform (an icebreaker) for two different years (2014 and 2016). I congratulate very much to the authors, and to the Canadian program NETCARE which should be an example of interdisciplinary studies to follow. I think the paper should be well accepted in ACP, following few major revisions which I am confident the author will be able to carry out.

The authors thank the referee for their thoughtful review of the manuscript and appreciate their kind words about the NETCARE program.

1) It is mentioned a number of times in the text that the Arctic marine microbial communities are likely to be responsible for the large increase of new particle formation recorded in 2016 vs 2014 (41% and 6% of the time, respectively). For example, on page 15 line 16-18 "The source strengths of gas-phase precursors and reactive species, which are generally not well understood in the marine environment, are key remaining factors for explaining differences in ultrafine particle production". Reading section 3.3, one may get the feeling that this is the main reason for the large increase observed in 2016 relative to 2014, given other meteorological and physical conditions did not change substantially. I suggest to modify the abstract (a bit too general in the current state) and report - for example - important conclusion such as line 33 pg 13 "CS

may not be a factor that directly limits the formation of UPF in this region". I think is important to stress that chemical precursors (likely coming from Arctic marine communities) may play an important rule in increasing UPF, and physical conditions (different CS, for example) may not be as important as chemical precursors availability. If that is the case, it should be stressed in the abstract, in the current stage too general and not representative of the discussions and conclusions presented across the manuscript.

Given the lack of a clear delineation of the importance of biogeochemical factors versus meteorological factors leading to the formation and growth of UFP within this study, the authors feel it is not prudent to change the abstract to reflect greater certainty in the conclusion of a biogenic driver behind inter-annual differences in UFP formation.  Indeed, our findings are suggestive that the differences in UFP formation and growth *may* be tied to differences in sea ice coverage and/or the phase of biological activity in the Canadian Arctic – and this study provides an interdisciplinary analysis of key observations. However, decreased cloud fraction and increased solar radiation were also documented in 2016 compared with 2014, and may also help lead to greater UFP formation and/or growth. Ongoing study within NETCARE using computational tools may help clarify conclusions about which factors may be driving UFP formation in this region. We have added a statement in the conclusions highlighting the importance of computational tools to deriving insight on the mechanisms behind UFP formation and growth.

Page 18, lines 18-20 – sentence added: "In addition, chemical transport models and other computational tools, in conjunction with the observations reported in this study, may lend important insight into the drivers of UFP formation in the Canadian Arctic, and may be translatable to other remote locations."

2) Figure S5 should be a main part of the paper (maybe as new Figure 12) because it stresses a major difference across the two different years (differences up to 13-25% in sea ice concentrations) - therefore associating UFP events to open water, higher percentages of sea ice marginal zones, and less packed ice. On this regards, the authors should refer to a recent paper (Arctic sea ice melt leads to new particle formation, Dall'Osto et al., 2017a, Scientific Reports | 7: 3318 | DOI:10.1038/s41598-017-03328-1), where air mass trajectory analysis and atmospheric nitrogen and sulphur tracers linked frequent nucleation events to biogenic precursors released by open water and melting sea ice Arctic regions. Additionally, when discussing this (I leave to the author if prefer to discuss this in the result discussion part or in the conclusion paper) they should also discuss this potential source (polar open water and sea ice marginal Arctic sea ice regions) and put into context another recent paper from NETCARE (Croft et al, 2016, Nature Comm, another possible source of UFP related to bird colonies).

The referee's suggestion to include Figure S5 in the main text (as Figure 12) has been included in the revisions. While a short discussion addressing the ability of discontinuous ice to enable air-sea interaction was present in the initial manuscript, references to the work by Dall'Osto et al. (2017) and Sharma et al. (2012) have now been included, and a further concluding statement at the end of the paragraph (p16, lines 2-5) has been added. The authors agree with the referee that the change in sea ice could lead to differences in air-sea exchange of volatile UFP precursors, and this has been clarified in the revised manuscript. A statement (p16, lines 18-21) connecting the discussion to coastal sources of UFP precursors has also been added, including not only bird colonies (Croft et al., 2016; Weber et al., 1998), but also intertidal zones (O'Dowd et al., 2002; Sipilä et al., 2016).  In addition, a discussion of the interaction of marginal sea ice and atmospheric chemistry in polar regions has now been added to the manuscript (p17, lines 21-29).

Minor comments

- Section 3.3.2 - oceanic conditions. I congratulate to the authors for improving the paper with this interdisciplinary part, an interesting one. Whilst DOC - among other marine biological measurements - during different years were almost identical, Figure S6 shows interesting differences for nitrate (among

other variables), suggesting phytoplankton production season was more advanced and well into post bloom phase during 2016 (relative to 2016). A recent paper (Dall'Osto et al., 2017, Antarctic sea ice is a source of organic nitrogen in aerosols, Scientific Report, DOI:10.1038/s41598-017-06188-x) also go in the same direction, showing sea ice marginal region with more advanced post bloom phase enhanced in UFP. It may be worth to stress that in polar regions (both Antarctic and Arctic) the biology is playing a role (and seems not Chl, but the stage of the bloom, is the key factor) and more interdisciplinary studies are needed.

As the referee describes, the manuscript discusses the importance of the bloom phase on marine biosphere-atmosphere interactions (e.g., p16 starting on line 10). The conclusion that the bloom state was different between the two summers was reached by holistically analyzing the nitrate concentration near the sea surface, primary production rates, dimethyl sulphide concentrations in seawater, and the nuanced information content within the size-segregated chlorophyll concentrations. The importance of post-bloom phase biogeochemistry on air-sea interactions has also been noted in various previous studies (Collins et al., 2013; Lee et al., 2015; O'Dowd et al., 2015; Wang et al., 2015), which are cited within the manuscript along with foundational studies of bloom dynamics (Azam et al., 1983). Indeed, biogeochemistry in the marginal ice zone is germane to this study as CCGS *Amundsen* encountered sea ice commonly during both summers, and sea ice concentrations in the region were a major distinguishing factor between the 2014 and 2016 expeditions. A discussion of the importance of the marginal ice zone to atmospheric chemistry in polar regions is warranted and has been added to the manuscript (p17, lines 21-29). It is important to point out that the impact of specific sources of UFP formation and growth precursors were not scrutinized in this study, as air mass histories indicate the inclusion of a variety of potential source characterizations (e.g., coastal, open water, sea ice) for each UFP formation event described in the manuscript. Consequently, a more broad approach to characterize the region as a whole was undertaken.

- pg 3 line 30-25. Whilst the authors do a decent job in addressing the different chemical precursors, it may be more appropriate to cite only works carried out in Arctic regions (not Atlantic or other oceans) and not forget Sippila et al 2016 (Nature) and also to address recent new findings (Croft et al., 2016, Birds colony emissions) and marginal sea ice (Dall'Osto et al., 2017a, Scien Rep).

Since so little is known about marine emissions of UFP precursors (and as the literature on aerosol nucleation chemistry develops in parallel), the authors feel that it is important to include relevant findings from various marine regions in light of the desire for process-level understanding in the long run. Relatively few measurements have been made to connect UFP formation with specific precursors, so it is important to keep an open mind on the possible chemical drivers. While it is becoming clear that organic material leads to particle growth in the Arctic (Willis et al., 2016), and that an accurate accounting for ammonia is important for modeling UFP formation (Croft et al., 2016), the level of understanding in the marine atmosphere lags behind the terrestrial biosphere to a great degree. The key points about sources of N-containing material in secondary marine aerosol and in biogenic material retrieved from sea ice are important contributors to accounting for possible marine UFP precursors (e.g., Dall'Osto et al., 2017; Facchini et al., 2008). As noted by the referee, UFP formation in coastal and marginal ice zones have been observed but much more information is needed to understand these processes fully. Iodine chemistry that is characteristic of intertidal emissions (Allan et al., 2015; Sipilä et al., 2016) and marginal ice emissions of sulfur- and nitrogen-containing compounds (Dall'Osto et al., 2017; Levasseur, 2013) are of interest to the community, and are now appropriately acknowledged and cited within the Introduction. (p3, lines 30-34)

- Pg 11 line 2-5, it is possible to access the importance of coastal vs open ocean sources? As Rev 1 suggests, is this study more related to a specific environment, such as Archipelagos, and not to be extrapolated to open ocean and marginal sea ice zone Arctic areas?

As noted in a prior comment to this referee (*vide supra*), the comparative analysis of marine biological parameters and atmospheric aerosol observations are deliberately broad in nature. Since the Canadian Arctic can have influences from coastal margins, open water regions, and transport from the high Arctic, the authors felt it prudent to retain a regional scale analysis that included all of these sources in a 'synoptic' sense. At the suggestion of the referees, a more developed discussion of the impacts of marginal ice, coastal seabirds, and intertidal zones has been included in various parts of Section 3 to clarify for the reader the potential importance of such sources to UFP formation and growth in this region. More targeted study of each of these sources may be required to understand their impacts on the environment as a whole.

- pg 15 line 23 - I think it is figure S5

This figure reference has been changed to 'Figure 12' as Figure S5 has been moved to the main text.

-pg 15 line 22. I think the authors should improve this section and decide what is more appropriate (if include figure S5 as figure 12, and expand this section). I think maybe presenting an average map for the two different seasons (2014 and 2016) but I am not sure the 1st of August is representative, I would use a longer period, or present figure S5 in the main text.

In accordance with another of this referee's comments, Figure S5 has been incorporated in the main text as Figure 12, which the authors feel enhances the manuscript in an important way. The spatial representation of Figure 11 remains unchanged in the revised manuscript, as it is provided as an example of the pattern of differences in sea ice concentration throughout the region, but the quantitative changes are well represented by Figure 12. In addition, the connection between differences in sea ice concentration and biological activity have been further developed in the context of UFP formation throughout Section 3.3.2, in response to the referee's comments. Discussions of interactions between aerosol chemistry and transport over open ocean (Sharma et al., 2012), associations between sea ice concentration and UFP formation (Dall´Osto et al., 2017), and possible influences of sea ice on marine biology in the surface ocean (Dall'Osto et al., 2017; Levasseur, 2013) have been enhanced or added to the manuscript.

10   Arctic locations have only recently been published (Asmi et al., 2016; Kolesar et al., 2017; Nguyen et al., 2016), and the present study is the first to present such an analysis strictly within the Arctic marine environment. The study is unique in its wide and consistent spatial coverage across two expeditions, the similar seasonal timing of the expeditions, and the large number of co-sampled atmospheric and oceanic parameters, permitting a wide range of environmental conditions to be considered. Specifically, these data are distinct from the more numerous Arctic UFP data sets that have been gathered at long-

15   term monitoring stations located on land.

The general characteristics of the atmospheric aerosol measured during each campaign will be described in detail and comparisons between expeditions will be made in light of their environmental similarities and differences. Further, in order to constrain the range of conditions that may limit UFP formation and growth in the Arctic marine environment, meteorological

20   and oceanic conditions were investigated throughout each of the expeditions. The goal of this study is to define the frequency and characteristics of UFP formation and growth within the remote Canadian Arctic, characterize the environmental factors that are associated with UFP formation in the Arctic marine boundary layer, and to provide broad motivation  for a more comprehensive understanding of  precursors to aerosol formation and growth in the marine/coastal environment.

25   **2    Measurement Methods**

**2.1    Atmospheric Aerosol Measurements**

Measurements of ambient atmospheric aerosol were conducted between 15 July – 12 August 2014 and 20 July – 23 August 2016 aboard the research icebreaker CCGS *Amundsen*, operating within the Canadian Arctic as part of a multi-year research project, NETCARE (Network on Climate and Aerosols: Addressing Key Uncertainties in Remote Canadian Environments).

30   The cruise track for each of the two field campaigns is provided in Figure 1.

Ambient concentrations of aerosol with $d_p > 4$ nm were measured using an ultrafine condensation particle counter (UCPC; TSI, Inc. Model 3776), operating with an inlet flow rate of 1.5 L min$^{-1}$. While the nominal lower size limit for detectable particles for this instrument was specified by the manufacturer at 2.5 nm, diffusional losses of particles in the tubing (stainless steel, 4.57 mm inner diameter) increased the practical lower size limit to approximately 4 nm. Concentrations were sampled at 1 Hz, and were subsequently averaged to time bins of coarser resolution for calculating various size-resolved aerosol metrics in conjunction with other data products. Number size distributions of particles between 10 – 430 nm were measured using a scanning mobility particle sizer (SMPS; TSI, Inc. Model 3080/3787) operating with a sample flow rate of 0.6 L min$^{-1}$ and a sheath air flow rate of 6.0 L min$^{-1}$. SMPS and UCPC sampled from the starboard side of the ship's foredeck, approximately 5 meters aft of the bow and approximately 7 meters above the sea surface. Number size distributions of particles with diameter between 0.54 – 20 μm were measured with an aerodynamic particle sizer (APS; TSI, Inc. Model 3021) from atop the ship's bridge using a louvered inlet designed for total suspended particle transmission and a straight vertical stainless steel tube (16.56 mm inner diameter) coupled directly to the inlet of the APS (total flow rate 5 L min$^{-1}$).

The influence of ship exhaust was excluded from the 2016 data by applying a wind direction filter to the data in post-processing with an acceptance angle of 60° to port and 90° to starboard of the ship's heading. Extension of the acceptance angle to 90° on the starboard side is related to the position of the sampling inlet near the starboard edge of the ship on the foredeck; winds arriving at the sampling inlet  directly perpendicular to the ship's heading from the starboard side were free from ship exhaust 
[revised manuscript text omitted]
., 2014). Long term measurements from Alert and Svalbard have shown associations between marine biogenic secondary aerosol tracers (sulphur and nitrogen compounds) and discontinuous sea ice in the Arctic (Dall'Osto et al., 2017b; Sharma et al., 2012). Recently, decreasing sea ice was also associated with UFP formation and growth (Dall'Osto et al., 2017b). In addition to remotely sensed sea ice coverage, the difference in the amount of time that CCGS *Amundsen* spent in sea ice between the two expeditions can also be noted in the histograms of sea surface temperature (SST; Figure 9c), where a distinct local maximum in normalized frequency of measurements just below 0 °C is notable only in 2014. Since discontinuous ice can lead to greater air-sea exchange of volatile precursors that have the demonstrated ability to form UFP, it is likely that the lower sea ice fraction in 2016 could have contributed to the difference in the frequency of UFP formation and growth observed in this study.

It has long been thought that secondary aerosol formation and its influence on CCN concentrations in the marine environment is associated with the atmospheric chemistry of marine biogenic organic and inorganic precursors  (e.g., Charlson et al., 1987; Clarke et al., 1998; Facchini et al., 2008; Fu et al., 2013; O'Dowd et al., 2002; O'Dowd and de Leeuw, 2007). Some assessments have tempered expectations of climate feedback mechanisms to exist in tropical and temperate zones (Heintzenberg et al., 2004; Pirjola et al., 2000; Quinn and Bates, 2011), but evidence for the formation of CCN-active secondary marine aerosol has been reported (Clarke et al., 1998; Dall'Osto et al., 2017b; Facchini et al., 2008; Willis et al., 2016). The prevalence of strong associations between UFP and air mass exposure to open ocean in the Arctic (Dall'Osto et al., 2017b; Sharma et al., 2012)  suggests that UFP formation and growth may be more strongly coupled to oceanic biological activity, since summertime conditions are highly favourable for UFP formation  (Heintzenberg et al., 2015; Leaitch et al., 2013; Leck and Bigg, 2010; Rempillo et al., 2011; Sharma et al., 2012). In addition, within the Canadian Arctic, the ratio of coastline to open water is relatively large compared to other regions, enhancing the importance of volatile precursor sources at land-ocean boundaries, like seabird colonies (Croft et al., 2016a; Weber et al., 1998) and intertidal zones (O'Dowd et al., 2002; Sipilä et al., 2016).

Emerging research suggests that ecosystem interactions within the surface ocean are more important to air-sea chemical interactions than the state of any single biological variable (Collins et al., 2013; Lee et al., 2015; O'Dowd et al., 2015; Wang et al., 2015). Oceanic measurements of both ecosystem interactions and biological state were analysed to assess the behaviour of microbiota in the surface ocean during the study period. Frequency distributions of different metrics that have been associated with marine microbiological activity are shown in Figure 9e-h: $DMS_{sw}$, DOC, chl *a*, and PP. While each of these metrics has a different relationship to the broad notion of 'marine microbiological activity', taken together, they may provide valuable information on the general state of the marine biological system during each expedition. Three of the four aforementioned metrics exhibited broadly similar characteristics between the two cruises, with subtle yet discernible,

differences. DOC concentrations were nearly identical between the two summers; it is possible that DOC concentrations in this region are driven to a substantial degree by physical processes (e.g., ocean mixing) and/or inputs from terrigenous sources (Dittmar and Kattner, 2003; Hansell et al., 2009) rather than just marine biological processes. While $DMS_{sw}$ concentrations have a similar modal concentration in the frequency distribution, a larger fraction of the measurements showed concentrations

5   greater than 1 nmol $L^{-1}$ in 2016 compared with 2014 (Figure 9e). In most locations, chl *a*, PP, and nitrate skewed lower in 2016 than 2014 (Figure 9g-i; see also Figure S5), suggesting that the phytoplankton production season was more advanced and well into the post-bloom phase. This difference is consistent with the greater contribution of small phytoplankton to chl *a* and may be related to the lower sea ice concentration in 2016. $DMS_{sw}$ production results from the enzymatic cleavage of the cellular osmolyte dimethylsulphoniopropionate (DMSP) often found in microalgae at different intracellular levels in relation

10   to phytoplankton species composition (Keller et al., 1989) and oxidative stress (Stefels et al., 2007; Sunda et al., 2002). The breakdown process from DMSP to DMS is performed by certain phytoplankton and is widespread among bacterioplankton (Stefels et al., 2007). Bacterial DMS production  may be more particularly important  when PP declines towards the later stages of a phytoplankton bloom (Azam et al., 1983). Changes in the chemical and physicochemical properties of marine aerosol have been associated with declining

15   phytoplankton abundance; it is thought that dynamic ecosystem processes, including hydrolytic enzyme interactions with organic matter, are important to aerosol production and composition (Collins et al., 2013; Lee et al., 2015; O'Dowd et al., 2015; Wang et al., 2015). Lower PP, nitrate, and chl *a* with higher $DMS_{sw}$ collectively suggest that the seasonal bloom was in a later phase of development in 2016 compared with 2014. which could  be associated with enhanced emissions of trace gases by

20   microbial communities (Carpenter et al., 2012; Schulz and Dickschat, 2007; Shaw et al., 2010), potentially including those that could act as precursors to UFP formation and/or aerosol growth. While the full suite of gases emitted from biologically active oceans is not well understood (although a dependence on community composition has been shown (e.g., Colomb et al., 2008)), the general understanding that trace gas production can be enhanced by certain biological interactions and productivity (e.g., Leck and Rodhe, 1991; Mäkelä et al., 2002; Shaw et al., 2010) points to an association with increased UFP formation

25   and/or growth, as observed in summer 2016.

It is plausible that the greater retreat of sea ice during 2016 compared with 2014 acted cooperatively with the observed differences in biological activity from the marine biogeochemical analysis provided above. Sea ice margins and under-ice biological communities have been suggested as important contributors to volatile sulphur- and nitrogen-containing precursors

30   to UFP formation in polar regions (Dall'Osto et al., 2017a; Levasseur, 2013). Indeed, UFP formation has been observed in the Arctic marginal ice zone and can be correlated with decreasing sea ice concentration over multi-year periods, although the factors driving Arctic UFP formation are still uncertain (Dall'Osto et al., 2017b; Karl et al., 2012; Leck and Bigg, 2010). The enhanced interaction of marginal or fragmented sea ice in 2016 may have contributed to the greater frequency of UFP formation and growth observed. At the same time, a greater retreat of sea ice could have also enhanced or changed the timing of the

seasonal phytoplankton bloom in the Canadian Arctic, causing the bloom to be in a more advanced developmental stage at the time of sampling in 2016 compared with 2014. Further study of biosphere-atmosphere interactions at or near the marginal ice zone and/or during the spring melt season may help elucidate mechanistic details that connect sea ice decay with UFP formation in polar regions.

**4    Conclusions**

[revised manuscript text omitted]

30  Clarke, A. D., Varner, J. L., Eisele, F., Mauldin, R. L., Tanner, D. and Litchy, M.: Particle production in the remote marine atmosphere: Cloud outflow and subsidence during ACE 1, J. Geophys. Res. Atmos., 103(D13), 16397–16409, doi:10.1029/97JD02987, 1998.

Collins, D. B., Ault, A. P., Moffet, R. C., Ruppel, M. J., Cuadra-Rodriguez, L. A., Guasco, T. L., Corrigan, C. E., Pedler, B. E., Azam, F., Aluwihare, L. I., Bertram, T. H., Roberts, G. C., Grassian, V. H. and Prather, K. A.: Impact of marine

biogeochemistry on the chemical mixing state and cloud forming ability of nascent sea spray aerosol, J. Geophys. Res. Atmos., 118(15), 8553–8565, doi:10.1002/jgrd.50598, 2013.

Colomb, A., Yassaa, N., Williams, J., Peeken, I. and Lochte, K.: Screening volatile organic compounds (VOCs) emissions from five marine phytoplankton species by head space gas chromatography/mass spectrometry (HS-GC/MS), J. Environ. Monit., 10, 325–330, doi:10.1039/b715312k, 2008.

Croft, B., Wentworth, G. R., Martin, R. V., Leaitch, W. R., Murphy, J. G., Murphy, B. N., Kodros, J. K., Abbatt, J. P. D. and Pierce, J. R.: Contribution of Arctic seabird-colony ammonia to atmospheric particles and cloud-albedo radiative effect, Nat. Commun., 7, 13444, doi:10.1038/ncomms13444, 2016a.

Croft, B., Martin, R. V., Leaitch, W. R., Tunved, P., Breider, T. J., D'Andrea, S. D. and Pierce, J. R.: Processes controlling the annual cycle of Arctic aerosol number and size distributions, Atmos. Chem. Phys., 16(6), 3665–3682, doi:10.5194/acp-16-3665-2016, 2016b.

Dal Maso, M., Kulmala, M., Lehtinen, K. E. J., Mäkelä, J. M., Aalto, P. and O'Dowd, C. D.: Condensation and coagulation sinks and formation of nucleation mode particles in coastal and boreal forest boundary layers, J. Geophys. Res., 107(D19), 8097, doi:10.1029/2001JD001053, 2002.

Dal Maso, M., Kulmala, M., Riipinen, I., Wagner, R., Hussein, T., Aalto, P. P. and Lehtinen, K. E. J.: Formation and growth of fresh atmospheric aerosols: Eight years of aerosol size distribution data from SMEAR II, Hyytiälä, Finland, Boreal Environ. Res., 10(5), 323–336, 2005.

Dall'Osto, M., Ovadnevaite, J., Paglione, M., Beddows, D. C. S., Ceburnis, D., Cree, C., Cortés, P., Zamanillo, M., Nunes, S. O., Pérez, G. L., Ortega-Retuerta, E., Emelianov, M., Vaqué, D., Marrasé, C., Estrada, M., Montserrat Sala, M., Vidal, M., Fitzsimons, M. F., Beale, R., Airs, R., Rinaldi, M., Decesari, S., Facchini, M. C., Harrison, R. M., O'Dowd, C. and Simó, R.: Antarctic sea ice region as a source of biogenic organic nitrogen in aerosols, Sci. Rep., 7, 6047, doi:10.1038/s41598-017-06188-x, 2017a.

Dall'Osto, M., Beddows, D. C. S., Tunved, P., Krejci, R., Ström, J., Hansson, H.-C., Yoon, Y. J., Park, K.-T., Becagli, S., Udisti, R., Onasch, T., O′Dowd, C. D., Simó, R. and Harrison, R. M.: Arctic sea ice melt leads to atmospheric new particle formation, Sci. Rep., 7, 3318, doi:10.1038/s41598-017-03328-1, 2017b.

[revised manuscript text omitted]

5 Sulfur in the Environment, ACS Symposium Series No. 393, edited by E. S. Saltzmann and W. J. Cooper, pp. 167–182., 1989.

Kivekäs, N., Carpman, J., Roldin, P., Leppä, J., O'Connor, E., Kristensson, A. and Asmi, E.: Coupling an aerosol box model with one-dimensional flow: a tool for understanding observations of new particle formation events, Tellus B, 68(0), doi:10.3402/tellusb.v68.29706, 2016.

Knap, A., Michaels, A., Close, A., Ducklow, H. and Dickson, A.: Protocols for the Joint Global Ocean Flux Study (JGOFS)

10 Core Measurements, in JGOFS Rep No. 19. Reprint of the IOC Manuals and Guides No. 29, p. 170, UNESCO Intergovernmental Oceanographic Commission, Bergen, Norway. [online] Available from: http://unesdoc.unesco.org/images/0009/000997/099739eo.pdf, 1996.

Kolesar, K. R., Cellini, J., Peterson, P. K., Jefferson, A., Tuch, T., Birmili, W., Wiedensohler, A. and Pratt, K. A.: Effect of Prudhoe Bay emissions on atmospheric aerosol growth events observed in Utqiaġvik (Barrow), Alaska, Atmos. Environ., 152,

15 146–155, doi:10.1016/j.atmosenv.2016.12.019, 2017.

Kreidenweis, S. ., Penner, J. ., Yin, F. and Seinfeld, J. .: The effects of dimethylsulfide upon marine aerosol concentrations, Atmos. Environ., 25(11), 2501–2511, doi:10.1016/0960-1686(91)90166-5, 1991.

Kulmala, M., Dal Maso, M., Mäkelä, J. M., Pirjola, L., Väkevä, M., Aalto, P., Miikkulainen, P., Hämeri, K. 
[revised manuscript text omitted]
. Note the difference in projection and spatial scale in the panels (e-h) to emphasize the local detail of the 0-200 m PES. Each of the release times corresponds to the observation of a UFP formation and/or growth event aboard CCGS *Amundsen*. Tracer release locations were dictated by the ship's coordinates at the given release time. Details on the model are described in Section 2.2.

[Figure]

**Figure 6: Time series plots of SMPS data from 2014 (top) and 2016 (bottom) NETCARE campaigns on CCGS *Amundsen*. The prevalence of ultrafine particle formation and growth events in 2016 is generally visible throughout the campaign and is in stark comparison with the more isolated nucleation and growth events observed in 2014.**

[Figure]

**Figure 7: Summary of parameters commonly relevant to ultrafine particle formation and growth in the atmosphere. In the top panel, filled circles represent the mean insolation (±1σ) for the period over which growth was measured, the horizontal dash marker is the maximum insolation for day on which the event began, and the horizontal grey dashed line is the average maximum incident shortwave radiation for days on which an event was *not* observed during each expedition. The second panel shows the condensation**

10 **sink (CS; ±1σ) during each period in which growth rate was measured. The third panel shows the particle size range over which**

**growth was observed/quantified. The bottom panel shows the apparent growth rate of particles (GR_app), measured as the rate of change in the modal diameter. The date and UTC time of the beginning of each event is coded within each event label, and the events are shaded to group by region.**

[Figure]

**Figure 8: (a.)** Normalized frequency distribution of the fractional contribution of particles measured by SMPS (CS_SMPS; $d_p$ = 10 – 430 nm) to the total CS (CS_TOT) during the 2016 campaign. **(b.)** Normalized frequency distribution of the fractional contribution of particles with $d_p$ < 100 nm (CS_100) to CS_TOT. **(c.)** Normalized cumulative frequency distribution of CS calculated from measured size distributions during the 2014 and 2016 campaigns. Coloured horizontal lines, shown for context, denote ranges of CS from the Canadian Arctic (this study; 1% and 99% values), a polluted continental site (Po Valley, Italy), a boreal forest site (Hyytiälä, Finland) (Westervelt et al., 2013), **a site on the northern coast of Alaska (Utqiaġvik [Barrow], summer)** (Kolesar et al., 2017), **Zeppelin station (near Ny Ålesund, Svalbard)** (Giamarelou et al., 2016), **and a coastal location in the North Atlantic (Mace Head, Ireland)** (Dal Maso et al., 2002). **The upper limit of the polluted continental site is off scale (~2×10^4 h^-1).**

[Figure]

**Figure 9: Normalized histograms of environmental parameters measured or sampled aboard CCGS *Amundsen* in 2014 and 2016.**
**(a.) Solar radiation, (b.) ambient air temperature, (c.) sea surface temperature, (d.) atmospheric relative humidity, (e.) dimethyl sulphide in surface seawater (DMS_SW), (f.) dissolved organic carbon (DOC) in surface seawater, (g.) chlorophyll-a (chl-a) in surface seawater captured on filters of two different porosities, (h.) primary productivity in surface seawater, and (i.) nitrate in seawater averaged over the top 35 meters of each depth profile.**

[Figure]

**Figure 10: MODIS-Aqua retrieval of monthly-averaged daily liquid cloud fraction for (a.) August 2014, (b.) August 2016, and (c.) the difference between 2016 and 2014. The dashed black box denotes the area within which the study-area average was computed. Data from NASA/GSFC on a Lambert Conformal Conic projection.**

[Figure]

**Figure 11: Sea ice concentrations within the Canadian Arctic, plotted for a single example day (1 August): (a.) 2014, (b.) 2016, and (c.) difference between 2016 and 2014. Data from NASA/NSIDC (Maslanik and Stroeve, 1999) on a Lambert Conformal Conic projection.**

Field Code Changed

[Figure]

**Figure 12: Time-resolved domain average of the difference in sea ice concentration between 2014 and 2016 within the Canadian Arctic (blue solid) and the domain average difference in sea ice normalized by the average 2014 concentration (red dashed; see Supplemental Material). Data from NASA/NSIDC (Maslanik and Stroeve, 1999).**

[revised manuscript text omitted]